# TAS3351 is a brain penetrable EGFR-TKI that overcomes T790M and C797S resistant mutations

Hidefumi Kasuga ✉, Yuki Kataoka, Fuyuki Yamamoto, Rei Miyamoto, Shingo Tsuji, Tatsuya Suzuki & Shinji Mizuarai

## Abstract

**Background** Activating mutations in the epidermal growth factor receptor (EGFR), particularly exon 19 deletions and L858R mutation, are frequently observed in non-small cell lung cancer (NSCLC) and confer sensitivity to EGFR-tyrosine kinase inhibitors (EGFR-TKIs). Among these EGFR-TKIs, osimertinib is currently the standard of care for patients with NSCLC harboring the activating mutations. However, resistant mutations often arise, leading to resistance to osimertinib. The resistant mutation that most frequently occurs in EGFR during osimertinib treatment is the C797S mutation. Another major resistant mutation arising in EGFR during treatment with other EGFR-TKIs, such as gefitinib and afatinib, is the T790M mutation. Currently, no approved EGFR-TKIs are effective in patients who simultaneously develop the T790M and C797S mutations. Additionally, brain metastasis often causes disease progression due to reduced drug penetration into the brain.
**Methods** We conducted preclinical evaluations of TAS3351, a fourth-generation EGFR-TKI, including biochemical, structural, and in vitro/in vivo pharmacological assays. We also evaluated the efflux transporter susceptibility and brain penetrability of TAS3351 in male mice.
**Results** Here, we demonstrate that TAS3351 overcomes resistance due to T790M and C797S mutations while sparing wild-type EGFR activity. Furthermore, TAS3351 is not a substrate of P-glycoprotein (P-gp) and the breast cancer-resistant protein (BCRP) and exhibits significant brain penetrability, resulting in anti-tumor efficacy in mice with intracranial allografts.
**Conclusions** These findings indicate that TAS3351 is a promising therapeutic candidate for patients with NSCLC whose tumors have relapsed or are refractory to treatment due to the C797S and T790M mutations, and the brain metastases.

## Plain language summary

Changes in a protein called the epidermal growth factor receptor (EGFR) are known to play a crucial role in the development of non-small cell lung cancer (NSCLC). Many drugs target EGFR, and these are called EGFR-tyrosine kinase inhibitors. Whilst they are often effective as a treatment for NSCLC initially, often the cancers develop resistance, which means that they stop responding to the treatment. Also, cancer often occurs in the brain, because many cancer drugs cannot reach the brain due to a barrier in blood transfer to the brain. Here we describe a EGFR-tyrosine kinase inhibitor that works against some of the cancers resistant to other EGFR-tyrosine kinase inhibitors and also can reach the brain effectively. These data suggest our drug, called TAS3351, has potential as an improved treatment option for patients with NSCLC.

Somatic mutations in the epidermal growth factor receptor (EGFR), especially exon 19 deletion (ex19del) and L858R mutation (both of which are commonly observed mutations in EGFR and collectively termed EGFR-activating mutations), are among the well-characterized oncogenic driver mutations in patients with non-small cell lung cancer (NSCLC)[1–3]. Patients with the ex19del or L858R mutations in EGFR account for approximately 30–50% of NSCLC cases in Asia, compared to 10–20% in Western countries[4–7]. EGFR is a member of the ErbB/HER family of tyrosine kinase receptors and plays an essential role in cell proliferation, differentiation, and survival. The tyrosine kinase activity of EGFR is tightly regulated by the

binding of its ligands (e.g., EGF) to its extracellular ligand-binding domain, resulting in the formation of homodimers or heterodimers with other ErbB/HER families and the mutual phosphorylation of these intracellular domains. EGFR-activating mutations cause constitutive activation of the catalytic reaction that phosphorylates its substrates, including the receptor itself, in a ligand-independent manner, thereby contributing to the development of NSCLC. Thus, much effort has been made to discover small molecules that inhibit the catalytic reaction of EGFR in an ATP-competitive manner, which are termed EGFR-tyrosine kinase inhibitors (TKIs). Presently, three major generations of EGFR-TKIs are broadly used in the

Discovery and Preclinical Research Division, Taiho Pharmaceutical Co., Ltd., Tsukuba, Japan. ✉e-mail: h-kasuga@taiho.co.jp

clinical setting: combined gefitinib[8], and erlotinib[9] as the first generation (reversible and selective to EGFR-activating mutations, sparing wild-type EGFR activity), combined afatinib[10], and dacomitinib[11] as the second generation (covalent and targeting the pan-ErbB/HER family), and osimertinib[12] as the third generation (covalent and selective to EGFR-activating mutations, sparing wild-type EGFR activity).

The first- and second-generation EGFR-TKIs initially demonstrated improved outcomes in patients with NSCLC harboring the EGFR-activating mutations. However, EGFR-TKI resistance is almost inevitable during treatment. Among EGFR-resistant mutations, the T790M mutation, in which the threonine residue at position 790 is substituted by methionine, has emerged as a secondary resistance mutation in approximately half of the patients treated with first- and second-generation EGFR-TKIs[13]. This mutation causes steric hindrance due to the bulkier side chain of methionine compared to that of threonine[14]. In addition, the T790M mutation increases the affinity of EGFR for ATP, thereby attenuating the effectiveness of ATP-competitive EGFR-TKIs compared to EGFR harboring the ex19del or L858R mutations[15]. Osimertinib was developed to overcome the resistance mediated by the T790M mutation through its covalent binding mechanism[16]. It has shown significant clinical benefits in patients with advanced NSCLC harboring the T790M mutation[17], as well as in EGFR-TKI treatment-naïve patients with NSCLC[12,18], establishing osimertinib as the standard of care for patients with advanced NSCLC harboring EGFR-activating mutations. Despite its superior clinical efficacy, large part of patients ultimately develop resistance to osimertinib[19,20]. The C797S mutation, in which the cysteine residue at position 797 is mutated to serine, occurs in approximately 10–20% of patients treated with osimertinib and is one of the most common mechanisms of EGFR on-target resistance[21]. Since cysteine 797 is the residue to which osimertinib covalently binds[16], osimertinib is no longer able to form a covalent bond with EGFR. However, as first-generation EGFR-TKIs reversibly interact with EGFR, preclinical studies have suggested that their inhibitory potency is not compromised by the presence of the C797S mutation[22,23]. Consequently, patients who develop C797S-mediated refractory tumors during first-line osimertinib treatment may respond to first-generation EGFR-TKIs. Nevertheless, the emergence of the T790M mutation during treatment with first-generation EGFR-TKIs after first-line osimertinib treatment will emerge as another clinical challenge. After developing a triple-mutation (ex19del/T790M/C797S or L858R/T790M/C797S) in EGFR, no approved EGFR-TKIs are currently available for treatment.

Since the discovery of the EGFR C797S mutation as a resistance mechanism to osimertinib, significant efforts have been made to develop next generation of EGFR-TKIs that can overcome the resistance caused by the T790M and C797S mutations. An early approach was EAI045, an allosteric inhibitor designed to overcome the resistance associated with C797S and T790M mutations while sparing wild-type EGFR activity[24]. This was followed by the development of derivatives such as JBJ-04-125-02[25] and JBJ-09-063[26]. The allosteric mechanism of action provides a rationale for combining these inhibitors with other ATP-competitive EGFR-TKIs, although they are ineffective against EGFR harboring an ex19del mutation. Another study focused on brigatinib, a TKI approved for the treatment of patients with anaplastic lymphoma kinase (ALK) mutations. Brigatinib, in combination with an anti-EGFR antibody, has demonstrated efficacy against tumors with EGFR-activating mutations along with T790M and C797S resistance mutations in a preclinical setting[22]. Based on these findings, a phase I/II clinical trial was initiated in 2020 (jRCT2031200231) to evaluate brigatinib plus panitumumab in patients with C797S-positive NSCLC following osimertinib treatment. However, the trial showed poor tolerability with a high incidence of early-onset pneumonitis[27]. BLU-945, developed by Blueprint Medicines, has emerged as the most advanced compound in clinical evaluation as a fourth-generation EGFR-TKI[28,29]. It is particularly notable for its high potency in the presence of the T790M mutation. BLU-945 has been evaluated as a monotherapy and in combination with osimertinib in patients with NSCLC. In heavily pretreated patients with NSCLC harboring EGFR-activating mutations, BLU-945 demonstrated clinical activity with acceptable tolerance.

However, owing to genomic heterogeneity, the responses were not durable. In contrast, the combination of BLU-945 and osimertinib exhibited clinical activity in patients who showed progression after osimertinib treatment. A correlation between the reduction in resistance mutation alleles detected by circulating tumor DNA (ctDNA) and tumor shrinkage was observed in the monotherapy and combination cohorts. Despite these promising results, Blueprint Medicines has announced the discontinuation of further development of BLU-945, citing the evolving external landscape and emerging clinical data.

The brain often serves as a sanctuary for metastatic tumors because of the blood-brain barrier (BBB), which restricts the entry of numerous anticancer agents into the brain parenchyma. This phenomenon is particularly pronounced in patients with NSCLC, who exhibit a higher incidence of brain metastases[30]. Therefore, there is a clinical need to develop agents capable of effectively penetrating the brain of patients with NSCLC. Notably, osimertinib has demonstrated superior brain penetration compared with other EGFR TKIs in preclinical models and has shown significant clinical benefits in patients with brain metastasis[12,18,31,32]. These findings suggest that the next-generation EGFR-TKI should exhibit enhanced brain penetration.

Here, we present TAS3351 as a fourth-generation EGFR-TKI, which is an orally bioavailable, brain-penetrant small-molecule inhibitor. Preclinical studies demonstrate that TAS3351 exhibits superior inhibitory activity against EGFR-activating mutations, with or without T790M and C797S mutations, compared with its activity against wild-type EGFR. These data highlight the clinical potential of TAS3351 as a therapeutic option for patients with NSCLC with EGFR-activating mutations, including those with T790M and/or C797S mutations, as well as central nerve system (CNS) metastases.

## Methods
### Materials
TAS3351 and Compound 1 were synthesized by Taiho Pharmaceutical Co., Ltd., and those chemical structures were confirmed using electrospray ionization mass spectrometry (ESI-MS) and nuclear magnetic resonance (NMR) spectroscopy. Gefitinib and dacomitinib were purchased from MedChem Express, and erlotinib, afatinib, and osimertinib were obtained from Carbosynth, LC Laboratories, and ChemScene, respectively. HCC827, NCI-H1975, A-431, NIH/3T3, and Caco-2 cell lines were purchased from the American Type Culture Collection (ATCC, cat# CRL-2868, CRL-5908, CRL-1555, CRL-1658, and HTB-37, respectively). Ba/F3 and PC-9 cell lines were provided by the RIKEN BioResource Center (BRC) through the National BioResource Project of the Ministry of Education, Culture, Sports, Science and Technology (MEXT), Japan (cat# RCB0805 and RCB4455, respectively). Jump-In GripTite HEK293 cells were purchased from Thermo Fisher Scientific (cat# A14150). Ba/F3, PC-9, PC-9 (C797S), HCC827, and NCI-H1975 cell lines were cultured in RPMI-1640 medium. Jump-In GripTite HEK293 cells, A-431, NIH/3T3, and Caco-2 cell lines were cultured in Dulbecco's Modified Eagle Medium (DMEM). All media were supplemented with 10% fetal bovine serum (FBS) and 1% penicillin-streptomycin. In addition, the following supplements were added in the medium in each cell line: Jump-In GripTite HEK293, 0.1 mmol/L non-essential amino acids (NEAA), 25 mmol/L HEPES, 600 μg/mL G418, and 200 μg/mL Hygromycin B, 10% dialyzed FBS instead of FBS; Ba/F3, 1 ng/mL mouse interleukin-3 (mIL-3), and 5 μg/mL puromycin; NIH/3T3-EGFR, 2 μg/mL puromycin; Caco-2, 2 mmol/L L-Glutamate, and NEAA. All cells were cultured and maintained at 37 °C and 5% $CO_2$. The genetic characteristics of all cell lines were confirmed using short tandem repeat (STR) profiling. Cell lines were confirmed to be free of mycoplasma using polymerase chain reaction (PCR) analysis before initiating the experiments.

### Generation of PC-9 C797S mutant cell line
PC-9 (C797S) cells were generated by gene editing using CRISPR-Cas9 to introduce the C797S mutation in EGFR into PC-9 cells. First, the gRNA complex was prepared by mixing Alt-R CRISPR-Cas9 crRNA and Alt-R tracerRNA, followed by incubation with Alt-R *S.p.* HiFi Cas9 Nuclease V3 at

room temperature for 10–20 min to prepare the ribonucleoprotein (RNP) complex. Next, the single-stranded donor oligonucleotides (ssODNs: 5'-ctg cct cac ctc cac cgt gca gct cat cac gca gct cat gcc ctt cgg ctc cct gct gga cta tgt ccg gga aca aga caa tat tgg ctc cca gta cct gct caa ctg gtg tgt gca-3') were mixed with the RNP complex, and the ssODN/RNP mixture was introduced to PC-9 cells by electroporation (Lonza). After electroporation, the PC-9 cells were cultured with the Alt-R HDR enhancer V2 and osimertinib for enhancing gene editing and selection of gene-edited cells. After selecting a single clone, the genomic sequence was confirmed to harbor the mutation causing the C797S mutation in EGFR. All reagents were purchased from Integrated DNA Technologies, unless otherwise specified.

### Generation of Ba/F3-EGFR and NIH/3T3-EGFR cell lines
Human full-length EGFR harboring a particular mutation or wild-type EGFR was cloned into the PB-CMV-MCS-EF1-RFP-T2A-Puro PiggyBac vector (System Biosciences) to construct the EGFR PiggyBac vector plasmids. Each EGFR PiggyBac plasmid was simultaneously transfected into Ba/F3 cells with Super PiggyBac Transposase Expression Vector (System Biosciences) using a Nucleofector (Lonza). Afterward, the cells were treated with puromycin to select the cells which were stably integrated EGFR. After the selection of a single clone for each EGFR mutant and wild-type, the EGFR sequence was confirmed to harbor a particular mutation or was a wild-type. To establish NIH/3T3 cell lines stably expressing both EGFR and luciferase, EGFR PiggyBac vectors and Super PiggyBac Transposase were simultaneously transfected into the cells using Lipofectamine 3000 according to the manufacturer's protocol (Thermo Fisher Scientific), followed by puromycin treatment to select the integrated cells. Thereafter, puromycin selected cells were transfected with luciferase-pJTI FAST and pJTI PhiC31 Int Vector (Thermo Fisher Scientific), followed by Hygromycin B treatment to select the integrated cells. After selecting a single clone for each EGFR mutant, the EGFR sequence was confirmed to harbor a particular mutation in EGFR.

### Off-chip mobility shift assay
The reaction mixtures were prepared by adding human recombinant EGFR (Carna Biosciences) to the reaction solution (the final concentration: 1 mmol/L Srctide, 1 mmol/L ATP, 5 mmol/L MgCl$_2$, and 1 mmol/L MnCl$_2$), 15 mmol/L HEPES, 0.0075% Triton X-100, 2.75 mmol/L DTT). The reaction mixtures and test compounds dissolved in dimethylsulfoxide (DMSO) were mixed and incubated for 1 hour at room temperature. The reaction was terminated by adding the Termination Buffer of QuickScout Screening Assist MSA (Carna Biosciences). The reaction mixture was applied to a LabChip system (PerkinElmer), and the product ($P$) and substrate ($S$) peptide peaks were separated and quantified. The kinase reaction was evaluated by calculating the product ratio from the peak heights of the $P$ and $S$ peptides ($P/(P + S)$). The readout value of the reaction control (complete reaction mixture) was set to 0% inhibition, the readout value of the background (without enzyme) was set to 100% inhibition, and the percentage inhibition of each test solution was calculated. IC$_{50}$ values were calculated from the concentration vs %inhibition curves by fitting to a four-parameter logistic curve using Microsoft Excel 2013 with the Solver add-in software.

### EGFR biochemical kinetics assays
EGFR biochemical kinetic parameters were calculated using the PhosphoSens Kinase Assay (Assay Quant Technologies) according to the manufacturer's protocol. Briefly, human recombinant EGFR (Carna Biosciences) was mixed with PhosphoSens Substrate and Reaction mix at various concentrations of ATP; thereafter, incubation was initiated at 30 °C, collecting fluorescence intensity (RFU) using EnVision 2105 (PerkinElmer) every 1 min for 1 h. The excitation and emission filters were 355 nm/40 nm-umbelliferone and 485 nm/14 nm-FITC (PerkinElmer), respectively. The initial velocity was evaluated using the slopes of the initial linear portion and fitted to the Michaelis-Menten equation to determine $V_{max}$ and $K_m$ values. $K_i^{app}$ for each test compound was determined using the same procedures at various concentrations of each test compound. Simultaneously, the

concentration of the active enzyme [E] of each recombinant human EGFR was determined using afatinib or TAS3351. $k_{cat}$ was estimated from $V_{max}$ and [E]. $K_i^{app}$ values were calculated according to Morrison plot. $K_i$ values were calculated using $K_i^{app}$, $K_{m[ATP]}$ values, and ATP concentration.

### Crystallography
Protein purification and crystallography were performed by Proteros Biostructures GmbH. Briefly, the kinase domains of wild-type (residues G696 to G1022) and mutated (residues G696 to G1022, T790M, L858R, E865A, E866A, and K867A were introduced) human EGFR were expressed in Sf9 cells. The proteins were purified using affinity chromatography and gel filtration to >95% purity as determined by Coomassie-stained SDS-PAGE. Purified proteins were concentrated to 8 mg/mL and used for crystallization study. The crystal of each EGFR was obtained at the following conditions: wild-type EGFR; 1.2 mol/L K/Na-tartrate, and 0.1 mol/L HEPES pH 7.0, mutated EGFR; 8.0% ethylene glycol, 0.1 mol/L HEPES pH 7.25, and 10% w/v PEG 10 K. EGFR-Compound 1 complex crystals were obtained by soaking each apo-crystal. X-ray diffraction data had been collected at the Swiss Light Source (SLS) using cryogenic conditions at final resolutions of 2.33 Å and 2.67 Å. The crystals belonged to the space group I23. Data was processed using the XDS and XSCALE programs. The PDB IDs are as follows: wild-type EGFR with Compound 1 (9KL4) and mutated EGFR with Compound 1 (9KLW).

### In-cell western assay
The inhibitory potency of the test compounds on phosphorylated EGFR (pEGFR) was evaluated by assessing the phosphorylated and total EGFR signals using Odyssey CLx (LI-COR). EGFR expression vectors for transient expression were constructed using pcDNA6.2/V5/GW (Thermo Fisher Scientific), followed by the confirmation of each EGFR sequence. To evaluate phosphorylation signals on transiently expressed EGFR, 10 μg of each EGFR expression vector was transfected into Jump-In GripTite HEK293 cells using ViaFect (Promega) according to the manufacturer's protocols; thereafter, the cells were cultured overnight. The PC-9, PC-9 (C797S), HCC827, NCI-H1975, and A-431 cell lines were plated in 96-well plates and cultured overnight. Each test compound was dissolved in DMSO at adjusted concentrations, and the cells were treated with the test compounds for 1 h. After the incubation, the cells were fixed with 20% Formalin Neutral Buffer Solution (FUJIFILM Wako Pure Chemical Corporation), and the wells were washed with phosphate buffered saline (PBS)-Tween 20, followed by treatment with 0.1% Triton-X 100 for membrane permeabilization and with Intercept (PBS) Blocking Buffer (LI-COR) for blocking of nonspecific binding. Both anti-pEGFR (Tyr1068, cat# 2234, Cell Signaling Technology) and anti-EGFR (cat# MA5-13697, Thermo Fisher Scientific) antibodies in Interceptor (PBS) Blocking Buffer (1:200 dilution, respectively) were added to the wells, and the plates were incubated overnight. The following day, the wells were washed with PBS-Tween 20, and both IRDye 800CW goat anti-rabbit IgG secondary antibody (cat# 926-32211, LI-COR) and IRDye 680RD goat anti-mouse IgG secondary antibody (cat# 926-68070, LI-COR) in Interceptor (PBS) Blocking Buffer (1:800 dilution, respectively) were added and incubated at room temperature for 1 hour, followed by washing the wells with PBS-Tween 20. Fluorescence intensity of 700 and 800 nm was scanned with Odyssey CLx, and the fluorescence intensity of the 800 nm was corrected by that of the 700 nm to obtain the signal ratio of pEGFR to total EGFR. IC$_{50}$ values were determined by nonlinear regression analysis (Marquardt algorithm; 2-parameter logistic model) using the SAS software (version 9.4) package with EXSUS (version 10.0.3).

### Cell viability assay
Cultured Ba/F3-EGFR cells were washed with PBS, suspended in RPMI-1640 medium supplemented with 10% FBS, and plated in 96-well plates. In Ba/F3-EGFR (wild-type) cells, hEGF (FUJIFILM Wako Pure Chemical Corporation) was added at a final concentration of 50 ng/mL. The PC-9, PC-9 (C797S), HCC827, NCI-H1975, and A-431 cell lines were harvested, plated in 96-well plates, and incubated overnight. The cells were treated with

appropriately diluted test compounds in DMSO and incubated for 3 days. Cell viability was analyzed using CellTiter-Glo 2.0 (Promega), according to the manufacturer's protocol. The $IC_{50}$ values were calculated by nonlinear regression analysis using XLfit. The $GI_{50}$ values were calculated using linear regression analysis in the SAS software package via EXSUS. The mean values from day 0 and 3 for the DMSO control and day 3 for the test compound-treated wells were used to determine the $GI_{50}$ values, which are the concentration that the test compound inhibits cell growth by 50% compared to the DMSO control on day 3.

### In vivo study of subcutaneous tumor transplanted mouse models

The experiments and procedures were approved by the Animal Experiment Committee and performed in accordance with the internal guidelines for animal experiments, enforced on April 1, 2013, at Taiho Pharmaceutical Co., Ltd. Five-week-old male nude mice (BALB/cAJcl-*nu/nu*, microbiological grade: Specific pathogen free, CLEA Japan) were subcutaneously implanted with expanded cells, which were suspended in PBS or Matrigel (Corning), into the lower back at a site slightly dorsal to the axilla. The tumor volumes were calculated using the following equation: Tumor Volume $(mm^3) = 0.5 \times$ major axis (mm) $\times$ minor axis (mm) $\times$ minor axis (mm). For in vivo efficacy studies, the mice were randomized into groups based on tumor volume when the average tumor volume reached approximately 100–200 $mm^3$. Tukey's test (SAS with EXSUS) was used to evaluate statistical significance ($p \geq 0.05$) on the logarithmically transformed mean tumor volume in each allocated group on Days 0 or 1, and no statistical significance was confirmed before initiating the experiments. During the experiments, the mice in each labeled cage were identified by the position of the cut on their pinnae. The test compounds were administered orally at a dose of 10 mL/kg. The tumor volume and body weight of each mouse were measured once every 2–5 days. The application of the humane endpoints was determined primarily based on tumor volume, body weight change, and systemic physical condition. For Pharmacodynamics Marker (PD) analysis, the mice were randomized into groups based on tumor volume when the average tumor volume reached approximately 400–600 $mm^3$, following the previously described procedure, including the identification of individual mice. After the oral administration of TAS3351, the mice were euthanized by cervical dislocation under isoflurane anesthesia at each time point, followed by tumor sampling. Tumor samples were excised and placed in homogenate tubes, frozen in liquid nitrogen, and stored in an ultra-low temperature freezer set at −80 °C until use. The vehicles used for each compound were as follows: TAS3351, 20% (2-hydroxypropyl)-β-cyclodextrin (HP-β-CD) and 0.1 mol/L HCl; erlotinib, 0.5% hydroxypropyl methylcellulose (HPMC) and 0.1% polysorbate 80; osimertinib, 0.5% HPMC and 0.1 mol/L HCl; afatinib, 0.5% methylcellulose and 0.4% polysorbate 80. Housing conditions were as follows: temperature, 20–26 °C; humidity, 30–70%; light cycle, 12 h-light/dark cycle; diet, CE-2 (sterilized by irradiation, CLEA Japan); drinking water, filtered and sodium hypochlorite-added water.

All animal experiments were conducted without blinding. To avoid potential sex-related variability and confounders, all animal experiments were performed exclusively in male mice, since cyclic fluctuation in sex hormones in female mice is expected to introduce additional physiological variability. Nevertheless, as the findings described herein are based solely on male mice, it is unclear whether these findings are also relevant to female mice. The number of mice in each group of each mouse model was determined based on previous experiments to ensure enough robustness while minimizing the number of animals used. To minimize potential confounders, each animal experiment was performed under similar conditions, regarding strain, age, sex, microbiological grade, housing conditions, and diet. Unless otherwise specified, all other animal experiments described elsewhere also adhered to these conditions.

### Immunoblotting analysis

Frozen tumor samples were homogenized in cold RIPA buffer (Thermo Fisher Scientific) supplemented with PhosSTOP (Merck KGaA) and

cOmplete, Mini, EDTA-free Protease Inhibitor Cocktail (Merck KGaA) using Multi-beads Shocker (Yasui Kikai Corporation). The tumor protein extracts obtained via centrifugation of the supernatants were stored on ice until the assay. The protein concentration of the extracts was determined using BCA Protein Assay (Thermo Fisher Scientific), according to the manufacturer's protocol. The protein concentration of the extracts was adjusted to 1 mg/mL using RIPA buffer and Lane Marker Reducing Sample Buffer (Thermo Fisher Scientific) to prepare SDS-PAGE samples; afterward, the samples were denatured by heating at 98 °C for 5 min. After cooling to room temperature, the extracts and samples were frozen until SDS-PAGE analysis. 5 μg/lane of each SDS-PAGE sample was subjected to SDS-PAGE analysis using 7.5% polyacrylamide gels (Bio-Rad Laboratories) and transferred to polyvinylidene difluoride (PVDF) membranes using a Trans-Blot Turbo Transfer System (Bio-Rad Laboratories). Membranes were blocked with the PVDF Blocking Reagent for Can Get Signal (TOYOBO). After blotting with primary antibodies, the membranes were incubated with horseradish peroxidase (HRP)-conjugated secondary antibodies. The antibodies were diluted in Can Get Signal Solution1/2 (TOYOBO) according to the manufacturer's protocol. Chemiluminescence signals were assessed using ECL Prime Western Blotting Detection Reagent and Amersham Imager 600 QC (Cytiva). pEGFR (Tyr1068; cat# 3777), EGFR (cat# 4267), pAkt (Ser473; cat# 4060), Akt (cat# 4691), pErk1/2 (Thr202/Tyr204; cat# 4370), Erk1/2 (cat# 4695), β-actin (cat# 8457), and rabbit IgG, HRP-linked (cat# 7074) were all purchased from Cell Signaling Technology. All antibodies were diluted to a concentration of 1:1000.

### P-glycoprotein (P-gp) and the breast cancer-resistant protein (BCRP) efflux transporter analysis

In vitro susceptibility of TAS3351 as a substrate for P-gp and BCRP was evaluated using Caco-2 cells. [³H]digoxin, [³H]estrone sulfate, and [¹⁴C]mannitol were purchased from PerkinElmer. Caco-2 cells were plated in 24-well multiwell insert systems (Corning) at $3.0 \times 10^4$ cells/insert and incubated for 21 days. On the day of experiment, the medium of each well was replaced with the receiver solution (Hanks' Balanced Salt Solution containing 4.2 mmol/L $NaHCO_3$, 10 mmol/L HEPES, and 0.2% DMSO adjusted to pH 7.4 using NaOH), and the plate was preincubated for 1 h at 37 °C. After preincubation, the receiver solution of each well was removed, and the donor solutions (containing 0.01–3 μmol/L TAS3351, 1 μmol/L [³H]digoxin, 0.1 μmol/L [³H]estrone sulfate, or 10 μmol/L [¹⁴C]mannitol for substrate, 0.2% DMSO) or the receiver solution was added to each well, and the plate was incubated for 2 h at 37 °C. After incubation, each receiver solution was collected, and the TAS3351 concentration in the receiver solution was determined using liquid chromatography tandem mass spectrometry (LC/MS/MS, SCIEX). The radioactivity of [³H]digoxin, [³H]estrone sulfate, and [¹⁴C]mannitol in the receiver solution was measured using a liquid scintillation counter (LSC, PerkinElmer). Radioactivity for data processing was calculated by subtracting the background radioactivity from the radioactivity measured in each sample. $P_{app}$ values of the test compounds across the Caco-2 cell monolayers were calculated from the amount of test compound permeated after 2-hour incubation.

### Assessment of brain penetrability in mice

Prior to the study, the animal use protocol was reviewed and approved by the Institutional Animal Care and Use Committee in Sekisui Medical Co., Ltd., and the study was conducted in accordance with the regulations and guidelines of animal experiments specified by the Testing Facility in Sekisui Medical Co., Ltd. On the day of administration, the mice were assigned to groups by the stratification body weight average method. TAS3351 was orally administered to intact male nude mice (BALB/cAJcl-*nu/nu*, 8 weeks old, CLEA Japan), and blood and brain samples were collected at 0.25, 0.5, 1, 2, 4, 6, 8, and 24 h post administration. For sampling at the designated time points, the mice were anesthetized by isoflurane inhalation and euthanized by exsanguination from the inferior vena cava using a heparin sodium-coated syringe equipped with an injection needle. After centrifugation of the blood samples, the collected plasma samples were stored in an ultra-low-

temperature freezer until the analysis. At the same time, the brain samples also collected, weighed, and mixed with PBS at ratio of 1:3 (w/v). After adding zirconia beads to the container, the brain samples were homogenized with Shake Master Auto (Biomedical Science) for 3 min to prepare the brain homogenate. The homogenate samples were stored in an ultra-low-temperature freezer until the analysis. The concentration of TAS3351 in plasma and brain homogenate samples was determined using LC/MS/MS. The areas under the concentration-time curve from 0 to 24 h ($AUC_{0\text{-}24\,hr}$) of plasma and brain were calculated from mean concentration values using Phoenix WinNonlin 8.1 (Certara). $K_{p,brain}$ was determined by dividing the total brain $AUC_{0\text{-}24\,hr}$ by the total plasma $AUC_{0\text{-}24\,hr}$. $K_{p,uu,brain}$ was determined by dividing the unbound brain $AUC_{0\text{-}24\,hr}$ by the unbound plasma $AUC_{0\text{-}24\,hr}$, and the unbound $AUC_{0\text{-}24\,hr}$ was determined by multiplying the total $AUC_{0\text{-}24\,hr}$ by the respective tissue unbound fraction. Housing condition was as follows: temperature, 20.0–26.0 °C; humidity, 30–70%; light cycle, 12 h-light/dark cycle; diet, MF (Oriental Yeast); drinking water, tap water.

### Protein binding assay

The in vitro unbound fractions of TAS3351 in the plasma and brain homogenates of nude mice were determined using Rapid Equilibrium Dialysis (RED) device (8 K MWCO, Thermo Fisher Scientific). Blank brain samples were homogenized in three volumes of PBS containing 50 mmol/L NaF. Plasma and brain homogenate were spiked with TAS3351 to achieve 0.5 μmol/L and 20 nmol/g as the final concentration, respectively. An aliquot of plasma or brain homogenate containing TAS3351 was added to the sample chamber of the RED device. An aliquot of PBS was then added to the buffer chamber of the same device. The plate containing plasma or brain homogenate and buffer was incubated at 37 °C for 8 h in a 5% $CO_2$ incubator with micro plate shaker. After incubation, samples were collected from the respective chambers and analyzed by LC/MS/MS. The unbound fractions in the plasma and brain homogenate were calculated as the ratio of the concentration of TAS3351 in the buffer chamber to that in the sample chamber of the RED device.

### In vivo study in intracranial tumor transplanted mouse models

NIH/3T3-EGFR (ex19del/T790M/C797S)-luc cells were suspended in PBS, diluted to $5.0 \times 10^6$ cells/mL, and stored on ice until implantation. Male nude mice (BALB/cAJcl-*nu/nu*, 6 weeks old, microbiological grade: Specific pathogen free, CLEA Japan) were each head-fixed under anesthesia with a stereotaxic apparatus, and 2 μL of the cell suspension was injected at 0.5 mm anterior, 2.0 mm right, and 3.5 mm deep from the bregma, whereby $1.0 \times 10^4$ cells per mouse were transplanted. The mice were anesthetized with an intraperitoneal injection of the MMB mixed agent, a saline with 0.075 mg/mL medetomidine hydrochloride (Nippon Zenyaku Kogyo CO., Ltd.), 0.4 mg/mL midazolam (Fuji Pharma Co., Ltd.), and 0.5 mg/mL butorphanol tartrate (Meiji Seika Pharma Co., Ltd.), at a volume of 10 mL/kg body weight. Propeto (Maruishi Pharmaceutical Co., Ltd.) was applied to the eyeballs to prevent drying during the transplantation. The day of cell transplantation was defined as day 0. 150 mg/kg Luciferin (Promega Corporation) was intraperitoneally administered to measure the level of luminescence (photons/s) using an IVIS Lumina II Imaging System (PerkinElmer) on days 8, 13, and 19. Mice were anesthetized with isoflurane and placed in the prone position in the IVIS Lumina II Imaging System chamber. Bioluminescent images were acquired 15 min after the luciferin administration. The total flux in the region of interest of the head was calculated using Living Image software (version 3.2.0.7800). Based on the total flux levels, the mice were randomized into three groups of ten mice each on day 8. No statistical significance ($p \geq 0.05$) between the logarithmically transformed mean total flux values in each group was confirmed by Tukey's test (SAS with EXSUS). Administration of the vehicle and TAS3351 started on the day after randomization (day 9) and continued until the day before the final evaluation of survival time. Survival time was defined as the time from day 0 until the predefined humane endpoints were applied or death. The application of the humane endpoints was determined primarily based on body weight change, behavior, and systemic physical

condition. Unless otherwise specified, the animal experiments and procedures followed those described in the previous sections.

### Statistics

The statistical analyses were performed with SAS (SAS Institute Inc., ver. 9.4) in an EXSUS (EPS Corporation, ver. 10.0.3) environment. In the studies of subcutaneous tumor-transplant mouse models, the difference in mean tumor volume between each treated group and the control group was analyzed using Dunnett's test. In the study of intracranial tumor transplanted mouse models, the difference of the mean logarithmically transformed total flux of each treated group with that of the control group was evaluated with Dunnett's test. Similarly, the difference in survival curves between the TAS3351 groups and the control group was evaluated using the log-rank test, which was implemented after confirming that the median survival time in the TAS3351 groups was longer than in the control group.

## Results
### Discovery and biochemical evaluation of TAS3351

To identify small molecules that potently inhibit EGFR harboring the activating mutations, regardless of the presence or absence of the T790M and C797S resistance mutations, we screened our internal chemical library, followed by conducting a structure-activity relationship (SAR)-guided medicinal chemistry campaign, and discovered TAS3351 (Fig. 1a). TAS3351 features a pyrrolopyrimidine-quinoline core structure similar to that of EGFR-TKIs previously reported by Taiho Pharmaceuticals (Supplementary Fig. 1; TAS-121[33], zipalertinib/TAS6417/CLN-081[34]). In an off-chip Mobility Shift Assay (MSA) using recombinant human wild-type and mutated EGFR, TAS3351 demonstrated a potent inhibition of EGFR enzymes, particularly in mutants with the EGFR-activating mutations combined with the C797S, T790M, or both under 1 mmol/L ATP concentration, which represents intracellular conditions[15] (Fig. 1b). In contrast, inhibition was completely abolished in EGFR harboring the T790M mutation when treated with erlotinib, and in EGFR with the C797S mutation when treated with osimertinib (Fig. 1b). At ATP concentrations near the $K_m$ values for ATP in each recombinant EGFR, where $IC_{50}$ values represent a direct measure of inhibitor-enzyme affinity according to the Cheng-Prusoff equation[35], TAS3351 exhibited $IC_{50}$ values in the sub-nanomolar range. This suggests that TAS3351 binds to the mutated EGFR with a high affinity and behaves as a tight-binding inhibitor (Supplementary Fig. 2).

To elucidate the interaction between TAS3351 and EGFR harboring the T790M mutation and investigate how the pyrrolopyrimidine-quinoline core structure contributes to this interaction, we analyzed the co-crystal structures of wild-type EGFR (Fig. 1c) and EGFR variants harboring T790M and L858R mutations (Fig. 1d) in complex with the TAS3351 analog, referred to as Compound 1 (Supplementary Fig. 1). Compound 1 interacted with wild-type EGFR by forming three hydrogen bonds between the NH of amino-pyrimidine and main-chain carbonyl group of Q791, the N3 of pyrimidine and main-chain amide group of M793, and the N of quinoline and side-chain hydroxyl group of T854 (Fig. 1c). Notably, the hydrogen-bonding network between Compound 1 and EGFR remained intact, regardless of the presence of the T790M mutation, suggesting that the compound effectively occupied the hydrophobic space, thereby enhancing its efficacy against EGFR harboring the T790M mutation (Fig. 1d). C797 was unlikely to be involved in the interaction between Compound 1 and either wild-type or mutant EGFR.

Next, we conducted a kinetic characterization of the recombinant EGFR proteins with the EGFR-activating and C797S mutations (Table 1). Initially, we measured the $K_m$ values of each recombinant EGFR against ATP. Both ex19del and L858R mutations resulted in an increase in mean $K_m$ values against ATP, indicating that EGFR-activating mutations lead to decreased affinity of EGFR for ATP. However, the T790M mutation restored ATP affinity to some extent in both ex19del/T790M and L858R/T790M mutations, consistent with previous findings[15,36,37]. Notably, our assays revealed no apparent increase in the $k_{cat}$ values for each mutated

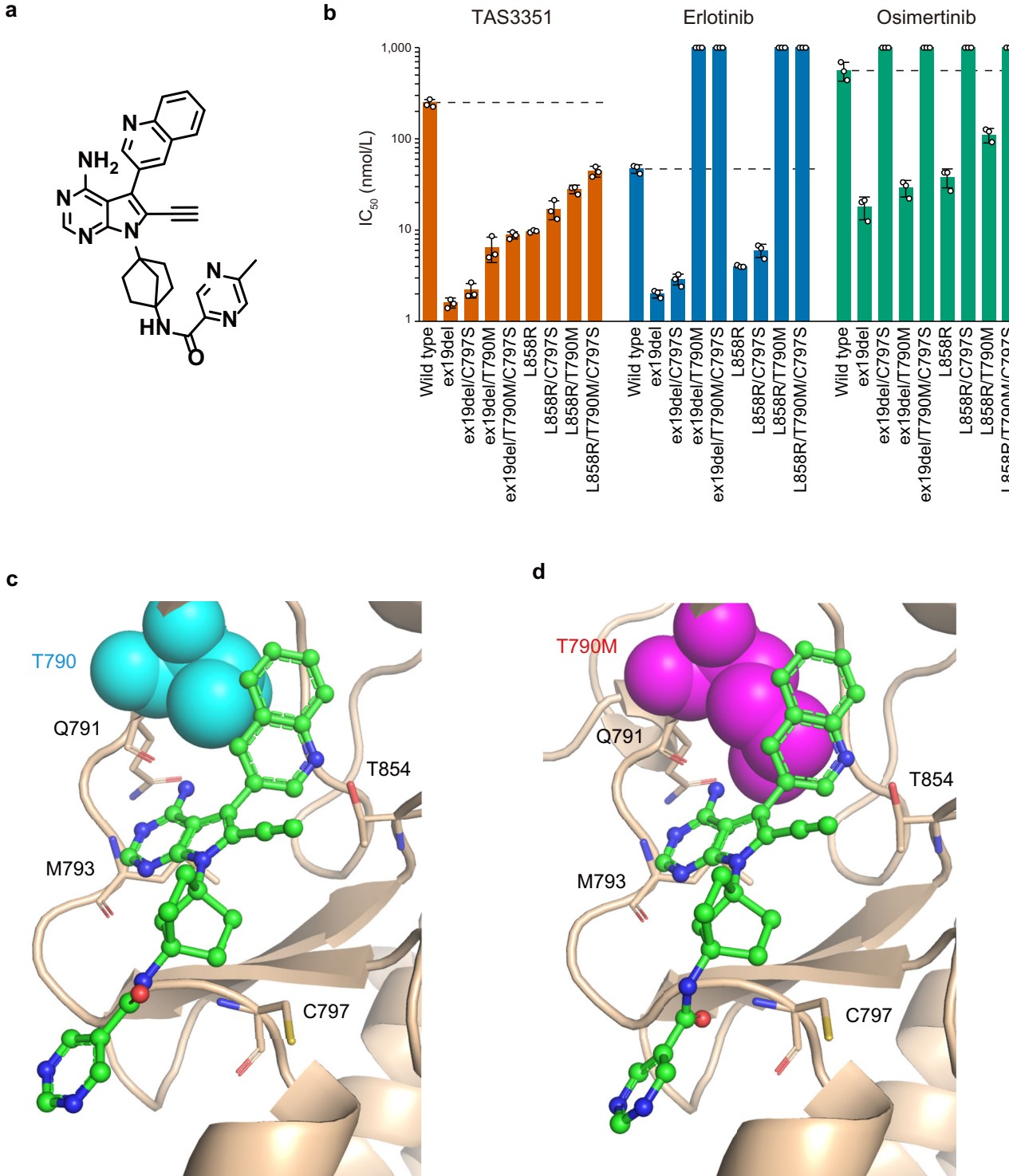

**Fig. 1 | Chemical structure of TAS3351, evaluation of EGFR enzyme activity, and co-crystal structures of EGFR with the TAS3351 analog. a** The chemical structure of TAS3351. **b** The mean $IC_{50}$ values of TAS3351, erlotinib, and osimertinib on recombinant EGFR enzyme activity. Enzyme inhibition was measured using the Off-chip Mobility Shift Assay (MSA) with recombinant human wild-type and mutant EGFR proteins in the presence of 1 mmol/L ATP. The dashed bars indicate the mean $IC_{50}$ values of each compound against wild-type EGFR. The $IC_{50}$ values in each protein were measured in triplicate and are depicted by individual dots. The bars indicate the mean $IC_{50}$ value in triplicate, and the error bars represent the standard deviation (s.d.) from the triplicate measurements. X-ray co-crystal structures of the wild-type human EGFR (**c**, PDB ID: 9KL4) and human EGFR harboring the T790M

and L858R mutations (**d**, PDB ID: 9KLM) complexed with Compound 1, a TAS3351 analog (chemical structure is shown in Supplementary Fig. 1). Compound 1 interacts with the wild-type and mutant EGFR by forming three hydrogen bonds with the main-chain carbonyl group of Q791, the main-chain amide group of M793 in the hinge region, and the side-chain hydroxyl group of T854. In addition, the quinoline group of Compound 1 tightly packs around the side chain of threonine. In the EGFR with T790M and L858R mutations (**d**), the side chain of methionine, which is bulkier than that of threonine, does not hinder the interaction with Compound 1 but rather fills the hydrophobic space more tightly than in wild-type EGFR. In both structures, C797 does not appear to be involved in the interaction between Compound 1 and EGFR.

**Table 1 | Enzyme kinetic parameters and inhibitory constants ($K_i$) of TAS3351 and erlotinib in wild-type and mutant recombinant EGFR**

| EGFR type | Kinetic parameters | | | | | Inhibitory constant ($K_i$) | | | |
|---|---|---|---|---|---|---|---|---|---|
| | $K_{m[ATP]}$[a] ($\mu$mol L⁻¹) | $V_{max}$[a] (nmol L⁻¹ m⁻¹) | $k_{cat}$[c] (s⁻¹) | $k_{cat}/K_{m[ATP]}$ (L mol⁻¹ s⁻¹) | [E][b] (nmol L⁻¹) | $K_i^{app}$[TAS3351][d] (nmol L⁻¹) | $K_{i[TAS3351]}$[e] (nmol L⁻¹) | $K_i^{app}$[erlotinib][d] (nmol L⁻¹) | $K_{i[erlotinib]}$[e] (nmol L⁻¹) |
| Wild-type | 29.7 ± 1.04 | 180 ± 13.6 | 3.97 | 1.34 × 10⁵ | 1.89 ± 0.0411 | 26.7 ± 6.09 | 5.30 | 4.42 ± 0.776 | 0.880 |
| Ex19del | 267 ± 10.2 | 165 ± 21.4 | 3.23 | 1.21 × 10⁴ | 2.84 ± 0.0578 | 0.0102 ± 0.00302 | 0.00480 | 0.290 ± 0.0205 | 0.137 |
| Ex19del/C797S | 314 ± 40.6 | 229 ± 11.4 | 6.42 | 2.05 × 10⁴ | 1.98 ± 0.416 | 0.0463 ± 0.0273 | 0.0237 | 0.321 ± 0.147 | 0.164 |
| Ex19del/T790M | 105 ± 4.19 | 142 ± 17.8 | 5.71 | 5.45 × 10⁴ | 1.38 ± 0.0754 | 0.0189 ± 0.00562 | 0.00649 | 4440 ± 782 | 1530 |
| Ex19del/T790M/C797S | 116 ± 1.86 | 163 ± 10.6 | 7.61 | 6.59 × 10⁴ | 1.19 ± 0.0727 | 0.0493 ± 0.00754 | 0.0180 | 7720 ± 1210 | 2830 |
| L858R | 395 ± 15.4 | 186 ± 13.3 | 7.55 | 1.91 × 10⁴ | 4.10 ± 0.150 | 0.338 ± 0.0743 | 0.259 | 0.267 ± 0.0446 | 0.205 |
| L858R/C797S | 515 ± 47.2 | 168 ± 8.30 | 11.3 | 2.20 × 10⁴ | 2.47 ± 0.144 | 1.21 ± 0.249 | 0.980 | 0.331 ± 0.0547 | 0.268 |
| L858R/T790M | 72.0 ± 6.75 | 169 ± 17.0 | 2.89 | 4.01 × 10⁴ | 3.25 ± 0.118 | 0.660 ± 0.477 | 0.224 | 5610 ± 1440 | 1900 |
| L858R/T790M/C797S | 67.3 ± 3.94 | 175 ± 11.4 | 2.07 | 3.07 × 10⁴ | 4.71 ± 0.0961 | 0.689 ± 0.341 | 0.224 | 4780 ± 1080 | 1550 |

EGFR epidermal growth factor receptor, *Ex19del* exon 19 deletions.

[a]$K_{m[ATP]}$ and $V_{max}$ were determined using the PhosphoSens Protein Kinase Assay with curve fitting of progress curves using the Michaelis-Menten model. The errors are standard deviations (s.d.) from triplicate measurements.

[b]Active enzyme concentration [E] was determined using afatinib or TAS3351. For EGFR harboring ex19del/C797S, ex19del/T790M, ex19del/T790M/C797S, and L858R/T790M/C797S, the initial velocity was plotted against each concentration of TAS3351, and the concentration of TAS3351 at which the initial velocity was expected to be zero is estimated as the value of [E] by linear regression. For the other EGFRs, afatinib was utilized in the similar procedure. The mean [E] was calculated from three independent experiments, and errors indicate s.d.

[c]$k_{cat}$ was estimated using the mean values of $V_{max}$ and [E].

[d]$K_i^{app}$ values were evaluated according to Morrison plot. The mean values and s.d of $K_i^{app}$ values from three independent experiments are displayed.

[e]$K_i$ values were calculated using $K_i^{app}$ and $K_{m[ATP]}$ values. Italic characters indicate enzymatic constants: $K_m$ (Michaelis constant), $V_{max}$ (Maximum Velocity), $k_{cat}$ (catalytic Constant), $K_i$ (Inhibitory constant), and $K_i^{app}$ (Apparent inhibitory constant).

EGFR compared to that of the wild-type EGFR; besides, the reaction efficiency of the mutated EGFRs, as calculated by the $k_{cat}/K_m$ ratio, remained below that of the wild-type EGFR. However, consistent with previous findings, we observed that the T790M mutation augmented the reaction efficiency of EGFR by 2- to 3-fold relative to each corresponding EGFR-activating mutation. The C797S mutation did not significantly affect $K_m$ values for ATP or $k_{cat}$ values, suggesting that, as expected, the C797 residue is not involved in the regulation of the catalytic activity of EGFR, and the C797S mutation does not confer any biochemical effect on EGFR enzyme kinetics. This is consistent with the fact that the intrinsic C797S mutation has not yet been detected, whereas the intrinsic T790M mutation has been detected[38,39]. Afterward, we calculated the $K_i$ values of TAS3351 for these recombinant EGFRs according to Morrison's equation[40], as TAS3351 is regarded as a tight-binding inhibitor, wherein the biochemical $IC_{50}$ values are dependent on the enzyme concentration[41,42]. The $K_i$ value of TAS3351 in EGFR with ex19del was determined to be 4.80 pmol/L, and this value was not changed significantly by the presence of C797S or T790M mutations. Conversely, the $K_i$ values of erlotinib, a non-covalent EGFR-TKI[43], increased in the presence of the T790M mutation, as previously reported[36], whereas the C797S mutation did not alter its $K_i$ values. Notably, TAS3351 exhibited larger $K_i$ values in EGFR harboring the L858R mutation (259 pmol/L) than in EGFR harboring the ex19del mutation (4.80 pmol/L), although the values in the L858R mutation were comparable to those observed with erlotinib (205 pmol/L). Similar trends were observed for $K_D$ values obtained from surface plasmon resonance (SPR) assay (Supplementary Fig. 3). Further analysis is required to elucidate the structural and biochemical basis to lead to the relatively large difference (over 50-fold) in the $K_i$ values of TAS3351 between ex19del and L858R mutations. Collectively, structural evaluations and biochemical analyses indicate that TAS3351 interacts effectively with EGFR harboring the T790M mutation, without being hindered by the methionine side chain in the T790M mutation or affected by the C797S mutation in EGFR.

Next, we investigated the kinome-wide selectivity of TAS3351 by assessing its inhibitory potency against 255 kinases at a concentration of 20 nmol/L. This analysis revealed that TAS3351 inhibited the enzymatic activity of 26 kinases by over 50%. Subsequently, we determined the $IC_{50}$ value of TAS3351 in these kinases, including EGFR with the ex19del mutation (Supplementary Fig 4.). Among them, TAS3351 most potently inhibited EGFR harboring the ex19del mutation, followed by seven additional kinases (HER4, EPHA5, EPHB4, EPHB3, EPHB2, RET, and YES), all of which demonstrated $IC_{50}$ values lower than those observed for wild-type EGFR. Given that the $K_i$ values of TAS3351 against EGFR harboring ex19del mutations were in the picomolar range, these results were sufficient to conclude that TAS3351 selectively inhibited EGFR with the ex19del mutation in the kinome. However, the potential secondary effects of the pharmacological inhibition of these eight kinases, with $IC_{50}$ values below that of the wild-type EGFR, remain to be elucidated.

## In vitro inhibitory potency of TAS3351 against phosphorylated EGFR and cell viability in genetically engineered models

We evaluated the inhibitory potency of TAS3351, along with other approved EGFR-TKIs, against in vitro phosphorylated EGFR (pEGFR) using an In-Cell Western assay, a quantitative, plate-based immunofluorescent assay[44]. Various mutant EGFRs, as well as the wild-type EGFR, were transiently expressed in HEK293-derived cells, and the pEGFR (Tyr1068) signal was quantified and normalized to the total EGFR signal (Fig. 2a, b). TAS3351 inhibited pEGFR with various mutations, including the T790M and C797S, as well as ex19del or L858R mutations. The mean $IC_{50}$ values ranged from 12.0 nmol/L for ex19del/C797S to 465 nmol/L for L858R/T790M/C797S, compared to 1150 nmol/L for wild-type EGFR. These findings suggest that TAS3351 selectively inhibits mutant EGFR activity regardless of the concurrent or sole resistant mutations of T790M and C797S, while sparing wild-type EGFR activity.

The inhibitory potency of osimertinib against EGFR was attenuated when EGFR harbored the C797S mutation, whereas its activity against

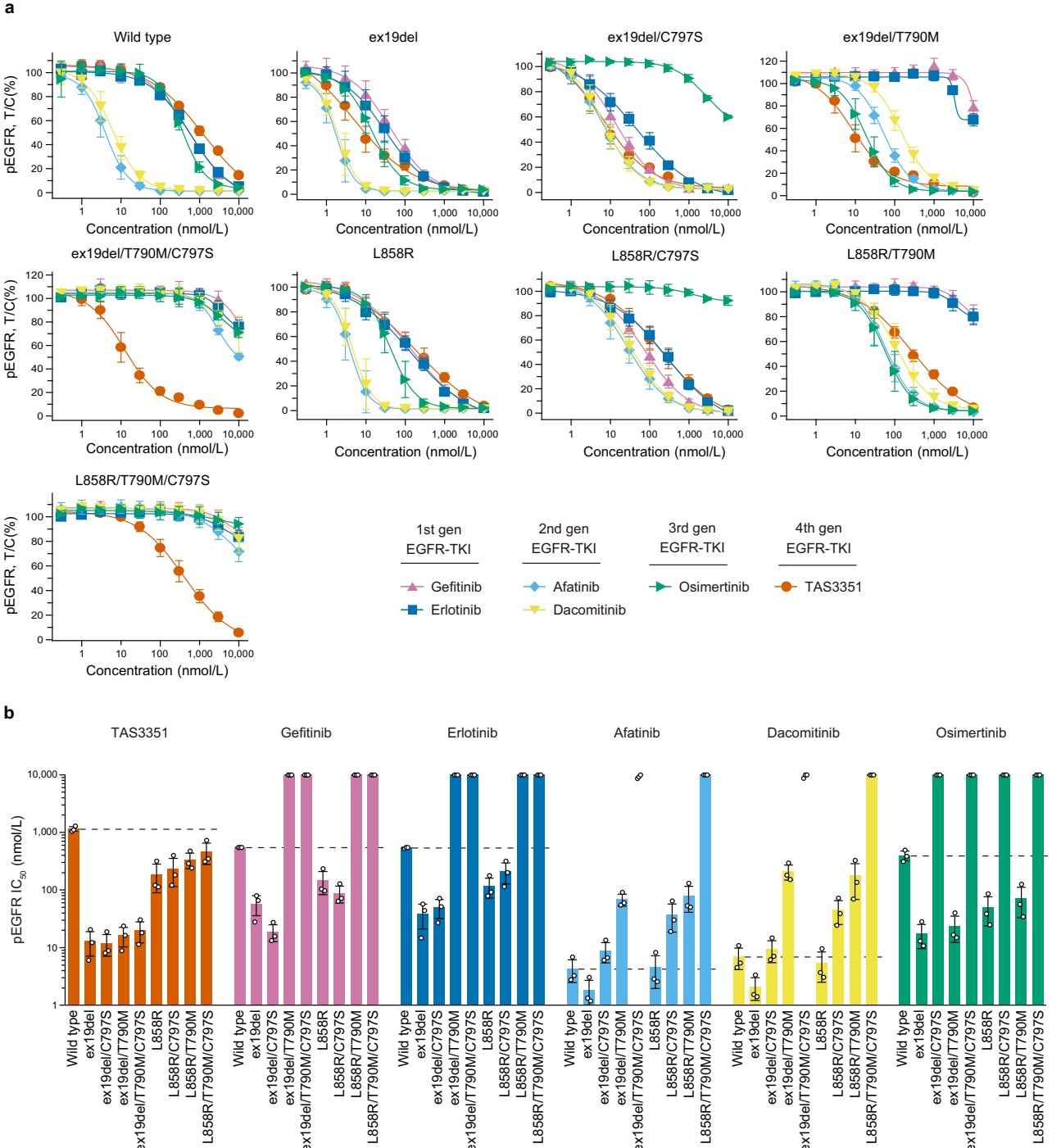

**Fig. 2 | In vitro inhibitory potency of TAS3351 and reference EGFR-TKIs against phosphorylated EGFR in transiently expressed EGFR. a** Wild-type and mutant EGFR were transiently expressed in Jump-In GripTite HEK293 cells, and phosphorylated EGFR (Tyr1068; pEGFR) was quantified by the In-Cell Western Assay. Total EGFR levels were measured simultaneously, and pEGFR signals were normalized with total EGFR levels, leading to the calculation of relative pEGFR levels. Cells were treated with TAS3351, reference EGFR-TKIs, and DMSO as control for 1 h. All experiments were conducted in triplicate, with compound concentrations titrated from 0.3 nmol/L to 10 μmol/L. Proportional levels of pEGFR at each concentration of the tested compounds are plotted as a percentage relative to the DMSO control (pEGFR T/C%, mean ± s.d.). All individual data are available in Supplementary Data 6. **b** The bars indicate the mean $IC_{50}$ values of TAS3351 and reference EGFR-TKIs in wild-type and mutant EGFR calculated from data shown in (**a**). The error bars represent s.d., and the individual $IC_{50}$ values are presented as dots. The dashed bars indicate the mean $IC_{50}$ values for wild-type EGFR for each tested compound. When the mean $IC_{50}$ values in triplicate were > 10 μmol/L, 10 μmol/L is presented as the mean $IC_{50}$ value. As the $IC_{50}$ values of afatinib and dacomitinib on EGFR with ex19del/T790M/C797S include both > 10 μmol/L and actual values, the mean $IC_{50}$ values are not calculated.

EGFR harboring the T790M mutation remained unaffected. Gefitinib and erlotinib exhibited weak inhibition of EGFR harboring the T790M mutation, whereas the C797S mutation did not significantly affect the potency. The inhibitory potency of afatinib and dacomitinib was partially attenuated by either T790M or C797S; however, the T790M/C797S concurrent mutation resulted in a substantial decrease in potency. As demonstrated in previous studies[45,46], afatinib and dacomitinib inhibited EGFR with the T790M mutation to some extent; nevertheless, their $IC_{50}$ values were higher than those observed for wild-type EGFR, resulting in insufficient potency for patients who develop the T790M mutation due to the dose-limiting toxicity

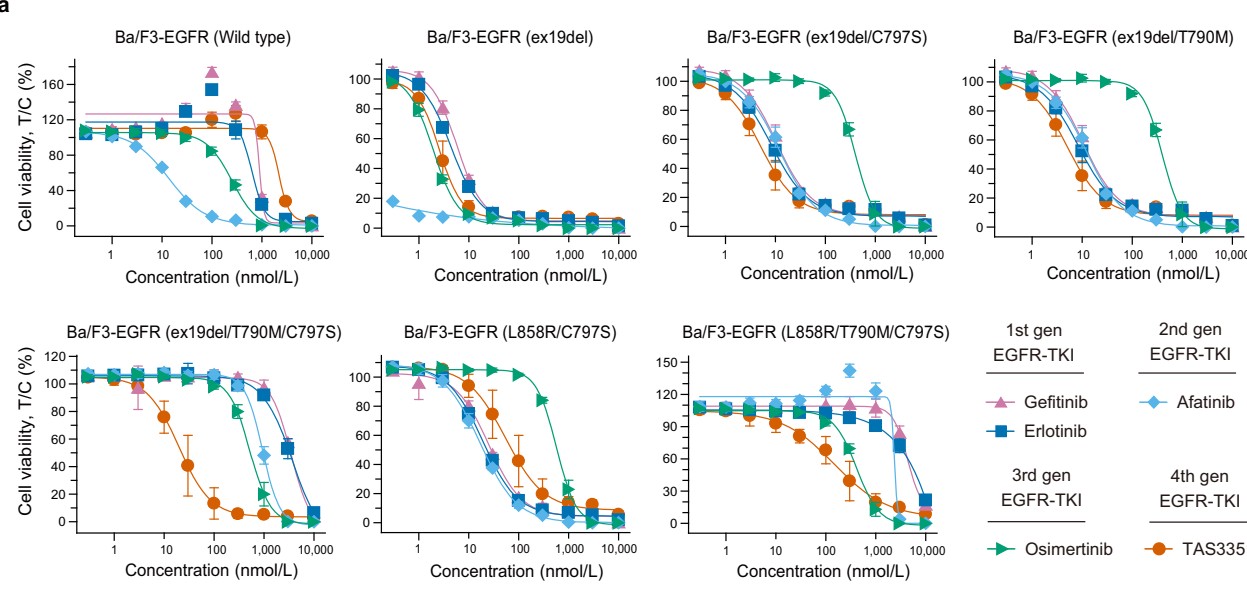

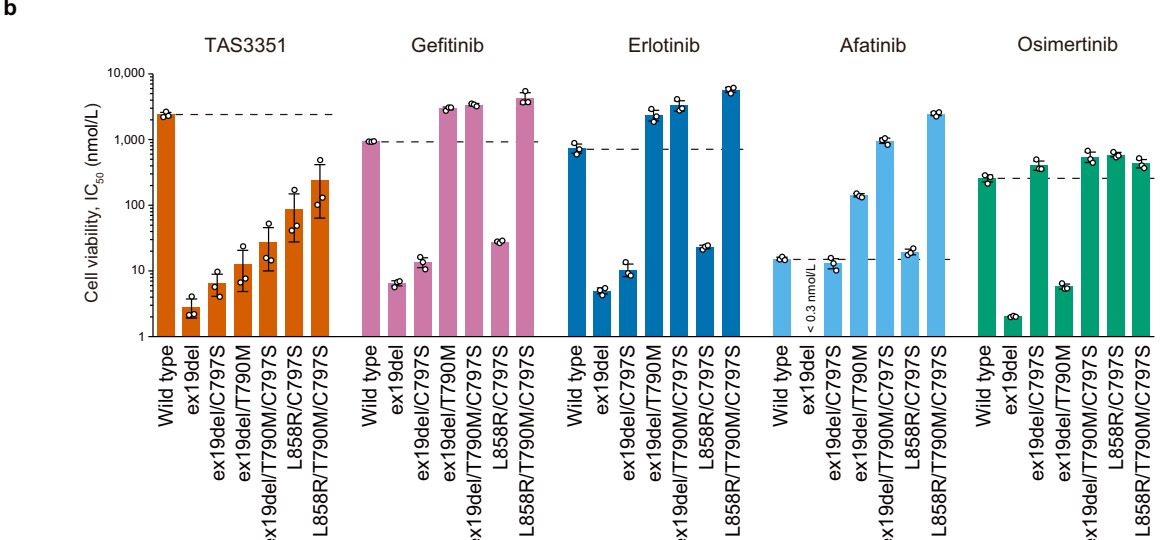

**Fig. 3 | In vitro inhibitory potency of TAS3351 and reference EGFR-TKIs against cell viability in Ba/F3-EGFR cells. a** The inhibitory potency of TAS3351 and reference EGFR-TKIs against cell viability was evaluated using murine Ba/F3 cells stably expressing wild-type or mutant human EGFR in the absence of mouse interleukin-3 (mIL-3). Human EGF was added to the culture medium for Ba/F3 cells expressing wild-type human EGFR. Cell viability was measured using the CellTiter-Glo assay. All compounds were tested using a range of concentrations (0.3 nmol/L to 10 μmol/L) for 3 days, and experiments were performed in triplicates. Results at each concentration are plotted as a percentage relative to the DMSO control (T/C%), with the mean values ± s.d. All individual data are available in Supplementary Data 7. **b** The bars indicate the mean $IC_{50}$ values calculated from data shown in (**a**), and the error bars indicate s.d. The individual $IC_{50}$ values are presented as dots. The dashed bars represent the mean $IC_{50}$ values of each compound for wild-type EGFR. When the mean $IC_{50}$ values in triplicate are > 10 μmol/L, 10 μmol/L is presented as the mean $IC_{50}$ value. The mean $IC_{50}$ value for afatinib is <0.3 nmol/L, as afatinib treatment in Ba/F3-EGFR cells harboring the ex19del mutation resulted in <50% cell viability at 0.3 nmol/L concentration compared to the DMSO control.

associated with the inhibition of wild-type EGFR[47,48]. Furthermore, the $K_i$ value of afatinib is comparable to that of erlotinib[36], indicating that afatinib can interact reversibly with EGFR harboring the C797S mutation. However, the reasons for the significant increase in the mean $IC_{50}$ values of afatinib and dacomitinib when EGFR acquires both T790M and C797S resistance mutations remain unclear.

Furthermore, we next evaluated the effect of TAS3351, in comparison to other EGFR-TKIs, on the viability of Ba/F3 cells, which were engineered to express either wild-type EGFR or various mutant EGFRs (Fig. 3a,b). Murine Ba/F3 cells require mouse interleukin-3 (mIL-3) for survival and proliferation; however, the introduction of mutated human EGFR enabled the transformation of Ba/F3 cells to ensure survival in an mIL-3-

independent manner[49]. Human wild-type EGFR allows Ba/F3 cells to survive in the presence of human EGF without mIL-3. Repeated attempts to introduce and transform Ba/F3 cells with human EGFR harboring the L858R or L858R/T790M mutations have been unsuccessful. TAS3351 exhibited inhibitory activity against EGFR with individual mutations, with the mean $IC_{50}$ values ranging from 2.84 nmol/L for the ex19del mutation to 241 nmol/L for the L858R/T790M/C797S mutation. These values were significantly lower than the mean $IC_{50}$ value observed in Ba/F3 cells expressing wild-type EGFR (2400 nmol/L). Meanwhile, the mean $IC_{50}$ values for gefitinib and erlotinib in Ba/F3 cells expressing EGFR with the T790M mutation were higher than those in cells expressing wild-type EGFR. Similarly, the mean $IC_{50}$ value of osimertinib in cells expressing

EGFR with the C797S mutation was higher than that in cells expressing wild-type EGFR. For afatinib, the mean $IC_{50}$ value for EGFR with ex19del was less than 0.3 nmol/L, which is lower than that for wild-type EGFR (15.4 nmol/L); however, the presence of either or both of C797S and T790M mutations resulted in comparable or higher mean $IC_{50}$ values than that for wild-type EGFR. Collectively, these findings indicate that TAS3351 effectively inhibits the cellular phosphorylation of EGFR harboring C797S with or without T790M mutations, leading to a reduction in cell viability while sparing the wild-type EGFR activity.

### Pharmacological activity of TAS3351 in human cancer cells intrinsically expressing mutated EGFR

In addition, we aimed to elucidate the pharmacological potency of TAS3351 in human cancer cell lines that intrinsically express mutated EGFR. Specifically, PC-9 and HCC827 cells harbor ex19del, and NCI-H1975 cells harbor L858R and T790M mutations. For comparison, A-431 cells expressing wild-type EGFR were evaluated. The PC-9 (C797S) cell line was generated by introducing a gene mutation into the EGFR genomic locus using CRISPR-Cas9-mediated gene editing, which enabled the expression of EGFR harboring the ex19del and C797S mutations. The inhibitory potency of TAS3351, erlotinib, and osimertinib was assessed by measuring pEGFR levels and cell growth (Fig. 4a, b). TAS3351 inhibited pEGFR, with the mean $IC_{50}$ values of 5.30–16.0 nmol/L in cells expressing mutated EGFR, whereas the mean $IC_{50}$ value in A-431 cells exceeded 10,000 nmol/L. The inhibition of cell growth by TAS3351 in these cell lines corresponded with pEGFR inhibition, yielding the mean $GI_{50}$ values of 2.93–21.9 nmol/L, compared to the mean $GI_{50}$ value of 120 nmol/L in A-431 cells. Erlotinib exhibited a trend similar to that of TAS3351 in inhibiting cellular pEGFR and cell growth, except in NCI-H1975 cells, in which the presence of the T790M mutation diminished its efficacy. Osimertinib potently inhibited both pEGFR and cell growth in cells expressing mutated EGFR, with the exception of PC-9 (C797S) cells. Overall, the inhibitory potency of cellular pEGFR and cell growth in the same cell lines was consistent with TAS3351 and other referenced EGFR-TKIs. These results in human cancer cell lines again underline that TAS3351 exerts selective inhibition in mutant EGFR regardless of T790M and C797S resistant mutations, with sparing wild-type EGFR activity.

### In vivo efficacy of TAS3351 in mouse models bearing NIH/3T3-EGFR allografts and xenografts with human cancer cells

To further elucidate the pharmacological effects of TAS3351, we established mouse tumor models bearing NIH/3T3 allografts that expressed human EGFR harboring the C797S mutation in combination with one of the EGFR-activating mutations, with or without the T790M mutation (Fig. 5a–d). NIH/3T3 mouse embryonic fibroblast cells are non-tumorigenic and form no obvious tumors when implanted subcutaneously in nude mice. However, NIH/3T3 cells expressing human EGFR harboring an activating mutation enable the formation of subcutaneous tumors in nude mice[50]. We selected NIH/3T3 models instead of Ba/F3 models for allograft assessment because the influence of mIL-3 in mouse serum could lead to confusing results in Ba/F3 allograft models. Once-daily oral administration of TAS3351 resulted in a statistically significant antitumor effect across all evaluated models (Fig. 5a–d), without apparent body weight loss (Supplementary Fig. 5). It is noteworthy that TAS3351 demonstrated comparable anti-tumor effects in models harboring mutant EGFR with the C797S resistant mutation alone (Fig. 5a, c) and in combination with the T790M resistant mutation (Fig. 5b, d), respectively. This is particularly significant given that the C797S mutation, in absence of T790M, is a major mechanism of on-target resistance to the first-line treatment of osimertinib in the current clinical setting. In contrast, once-daily oral administration of 25 mg/kg osimertinib, which achieved equivalent exposure to the clinically recommended dose of 80 mg[32], demonstrated minimal efficacy in all evaluated models. Conversely, oral administration of 80 mg/kg erlotinib produced a robust antitumor effect in models expressing EGFR with ex19del/C797S or L858R/C797S mutations (Fig. 5a, c). However, erlotinib exhibited reduced potency or no

significant effect in the models expressing EGFR with the T790M mutation (Fig. 5b, d).

In addition to NIH/3T3-EGFR allograft models, we evaluated TAS3351 in xenograft models with human cancer cell lines previously assessed in vitro (Fig. 4). Consistent with the inhibitory trends observed in vitro studies, TAS3351 exhibited statistically significant antitumor effect in xenograft models expressing mutant EGFR (Fig. 5e–h). In the A-431 xenograft model (Fig. 5i), afatinib was used as the positive control for wild-type EGFR inhibition. Notably, treatment with erlotinib, osimertinib, and afatinib all led to a statistically significant antitumor effect. However, TAS3351 did not demonstrate statistically significant inhibition of xenograft growth, at least at doses below 50 mg/kg. The $p$-value for the 80 mg/kg TAS3351 treatment group was 0.0350. Although A-431 cells are frequently employed as models to evaluate wild-type EGFR activity, extensive EGFR amplification and addiction have been reported[51]. Taken together with the findings from the NIH/3T3-EGFR allograft models, TAS3351 demonstrates robust antitumor potency in vivo at tolerable doses in mice, with similar trends to those observed in the previous in vitro studies.

To further investigate the correlation between target inhibition and antitumor efficacy, we conducted an in vivo pharmacodynamic (PD) marker analysis using a mouse xenograft model with the NCI-H1975 cell line (Fig. 6a, b). Male nude mice were orally administered TAS3351 at doses of 40 and 80 mg/kg, and tumor samples were collected 1, 4, 8, 16, and 24 h after administration. We evaluated the phosphorylation and protein levels of EGFR and its downstream signaling proteins, Akt and Erk1/2. Notably, the phosphorylation of EGFR, Akt, and Erk1/2 was significantly reduced in both the 40 and 80 mg/kg TAS3351 groups compared to the control group during the 1–8 h post-administration window. However, these phosphorylation levels started to recover 16 h after administration. By 24 h, the phosphorylation of these proteins was partially inhibited in the 80 mg/kg TAS3351 group, whereas the levels in the 40 mg/kg group were restored to nearly baseline, similar to the control group. The TAS3351 treatment reduced the levels of phosphorylated EGFR, as well as its downstream proteins, phosphorylated Akt and Erk1/2, in a largely parallel manner. Given the dose-dependent antitumor efficacy of TAS3351 in NCI-H1975 xenograft model (Fig. 5h), these results of PD marker analysis suggest that the degree of continuous PD marker inhibition by TAS3351 contributes to its in vivo efficacy in mice. Furthermore, it would be of great interest to determine whether split dosing (e.g., twice per day) improves anti-tumor efficacy by achieving continuous PD marker inhibition.

### Assessment of TAS3351 as a substrate for P-gp and BCRP efflux transporters and brain penetrability

P-gp and BCRP are well-characterized efflux transporters expressed in the BBB and involved in the elimination of various agents from the CNS[52]. Therefore, for a compound to effectively penetrate the brain, it should ideally circumvent being a substrate for these efflux transporters. Thus, we assessed the susceptibility of TAS3351 as a substrate for both P-gp and BCRP in vitro using human colorectal adenocarcinoma-derived Caco-2 cells, which endogenously express P-gp and BCRP and are widely used for the efflux transporter analysis in drug discovery[53]. The apparent permeability coefficient ($P_{app}$) values were determined by measuring the transport of the test compounds across the cell monolayers in the apical to basal and basal to apical directions, followed by the determination of $P_{app}$ ratios by dividing the $P_{app}$ values of basal to apical by those of apical to basal. If the $P_{app}$ ratio of the compound was higher than 2, then it was considered a potential P-gp or BCRP substrate[54]. The $P_{app}$ ratios of TAS3351 were 0.8–1.2 across various concentrations, indicating a similar level of $P_{app}$ values in both directions (Fig. 7a). In addition, [³H]digoxin and [³H]estrone sulfate, positive controls for each P-gp and BCRP, exhibited $P_{app}$ ratios of 9.6 and 18.7, respectively. The $P_{app}$ values of [¹⁴C]mannitol, a positive control for paracellular permeability, were $0.612 \times 10^{-6}$ cm/s from apical to basal direction and $0.461 \times 10^{-6}$ cm/s from basal to apical direction, suggesting that the assay was valid for assessing P-gp and BCRP substrates since almost no paracellular permeability was observed. Therefore, these data suggest that TAS3351 is not a P-gp or BCRP substrate.

**a**

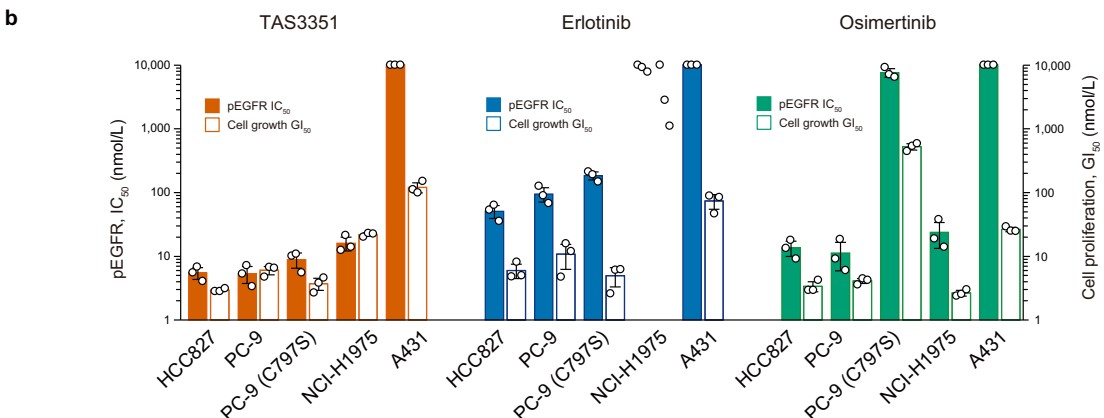

**b**

The brain penetrability of TAS3351 in mice was evaluated by assessing the unbound plasma and brain concentrations of TAS3351 after a single oral administration to male nude mice at doses of 20, 40, and 80 mg/kg, which were determined based on in vivo mouse efficacy studies. The plasma and brain concentration-time profiles over 24 hours after a single oral administration in nude mice demonstrated that the concentrations of TAS3351 in

the brain were comparable to or moderately lower than those in plasma (Fig. 7b). The mean unbound fraction of TAS3351 in the plasma (0.08%) and brain (0.17%) was determined by the rapid equilibrium dialysis (Fig. 7c), followed by determination of the total and unbound brain-to-plasma AUC ratios ($K_{p,brain}$ and $K_{p,uu,brain}$, respectively) of TAS3351 at doses of 20, 40, and 80 mg/kg (Fig. 7d). The $K_{p,uu,brain}$ value is considered a major

**Fig. 4 | In vitro inhibitory potency of TAS3351, erlotinib, and osimertinib against phosphorylated EGFR and cell proliferation in human cancer cell lines. a** Effects of TAS3351, erlotinib, and osimertinib on EGFR phosphorylation and cell proliferation were evaluated in human cancer cell lines. Each compound was tested at concentrations ranging from 0.3 nmol/L to 10 μmol/L. Phosphorylated EGFR (pEGFR) was assessed using the In-Cell Western assay with 1-h treatment with the compounds. Results are plotted as the percentage values relative to the DMSO control (T/C, %) on day 0 (0%) and day 3 (100%), with the mean values and s.d. obtained from triplicate experiments. Cell proliferation was measured by assessing cell viability on day 0 and day 3 in DMSO control wells and on day 3 in the test compound-treated wells using the CellTiter-Glo assay. All individual data are available in Supplementary Data 8. **b** The bars indicate the mean $IC_{50}$ values for pEGFR and the mean $GI_{50}$ values for cell proliferation in the human cancer cell lines, which are calculated based on the data presented in (**a**). The $GI_{50}$ values indicate the concentration of the test compound where the cell proliferation is inhibited by 50% compared to the DMSO control on day 3. The error bars represent s.d. ($n = 3$). The dots indicate individual $IC_{50}$ and $GI_{50}$ values. When all $IC_{50}$ values in a triplicate experiment exceeded 10 μmol/L, the mean $IC_{50}$ value is reported as 10 μmol/L. As the $IC_{50}$ and the $GI_{50}$ values of erlotinib on NCI-H1975 include both > 10 μmol/L and actual values; the mean $IC_{50}$ and $GI_{50}$ values are not calculated.

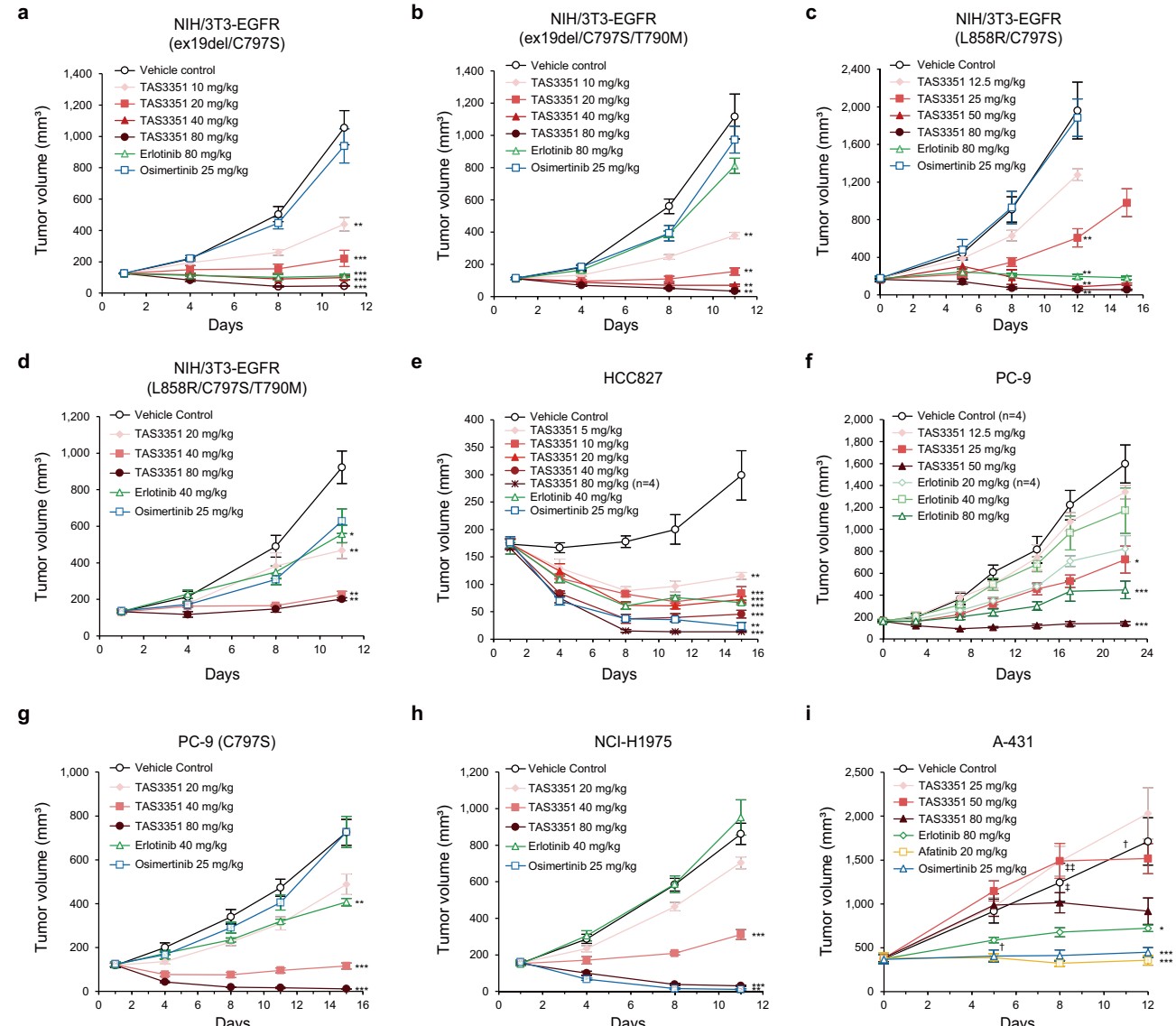

**Fig. 5 | In vivo efficacy of TAS3351 and reference EGFR-TKIs in mouse models bearing allografts of NIH/3T3-EGFR cells and xenografts of human cancer cells.** The in vivo efficacy of TAS3351, erlotinib, and osimertinib was evaluated in nude mice (BALB/cAJcl-nu/nu) subcutaneously implanted with murine NIH/3T3 embryonic fibroblast cells stably expressing human EGFR with the following mutations: ex19del/C797S (**a**), ex19del/C797S/T790M (**b**), L858R/C797S (**c**), and L858R/T790M/C797S (**d**). The in vivo efficacy of TAS3351 and reference EGFR-TKIs was evaluated in mice subcutaneously bearing xenografts of human cancer cell lines. The EGFR expressed in the respective human cancer cell lines is as follows: HCC827 (ex19del, **e**), PC-9 (ex19del, **f**), PC-9 (C797S) (ex19del/C797S, **g**), NCI-H1975 (L858R/T790M, **h**), and A-431 (wild-type, **i**). Number of mice in each group was five, except for PC-9 (C797S), in which the number was six, unless otherwise specified. Based on the predefined humane endpoint criteria for this study, [†]each animal in the vehicle control and erlotinib group was euthanized under anesthesia on day 11 and day 5, respectively, due to ulceration in the tumors, and [‡]an animal in the vehicle control and two animals in the TAS3351 50 mg/kg group were euthanized under anesthesia on day 8 due to exceeded tumor volume. The mouse in the 80 mg/kg TAS3351 group in HCC827, and the mouse in the vehicle control and 20 mg/kg erlotinib group in PC-9 were excluded from the evaluation, based on the predefined exclusion criteria (administration error). The mean tumor volumes are plotted, and the error bars represent the standard error of the mean (s.e.m.). All individual data are available in Supplementary Data 6. All compounds were administered orally once daily (QD) starting from day 1. In the control group, the vehicle of TAS3351 (20% HP-β-CD, 0.1 mol/L HCl) was administered as a control. Statistical analyses are performed with Dunnett's test to compare the logarithmically transformed tumor volume on the same day of each treated group with that of the control group. *$p < 0.01$, **$p < 0.001$, ***$p < 0.0001$.

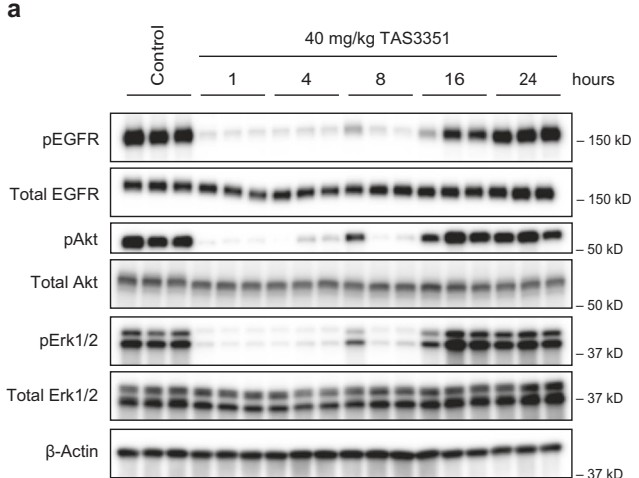

**Fig. 6 | Evaluation of pharmacodynamic markers by a single administration of TAS3351 in nude mouse bearing NCI-H1975 xenografts.** TAS3351 was orally administered at 40 mg/kg (**a**) and 80 mg/kg (**b**) doses to nude mice (BALB/cAJcl-nu/nu) bearing subcutaneous NCI-H1975 xenograft tumors. Tumors were collected at 1, 4, 8, 16, and 24 h after administration. The control group received the vehicle of TAS3351 (20% HP-β-CD, 0.1 mol/L HCl). Each group consisted of three mice. The pharmacodynamic (PD) markers in the tumor samples were analyzed by immunoblotting analysis. The PD markers analyzed in this study were phosphorylated EGFR (pEGFR, Tyr1068), phosphorylated Akt (pAkt, Ser473), and phosphorylated Erk1/2 (pErk1/2, Thr202/Tyr204). β-actin was evaluated as a loading control.

indicator of the pharmacological activity in the CNS, according to the free drug hypothesis, where the free drug concentration, rather than the total drug concentration, determines the activity[55,56]. The $K_{p,uu,brain}$ values for TAS3351 were greater than 0.3 at all doses and greater than 1.0 at 40 and 80 mg/kg doses, indicating that TAS3351 has a remarkable brain penetrability.

### Antitumor activity of TAS3351 in an intracranially transplanted mouse allograft model

To determine whether the remarkable brain penetrability of TAS3351 contributes to its therapeutic efficacy against CNS tumors, we established a mouse allograft model by intracranial transplantation of NIH/3T3 cells expressing human EGFR with the ex19del/T790M/C797S mutation together with luciferase (Fig. 8a–d). Mice with intracranial allografts exhibited a robust increase in luciferase activity and a rapid decrease in survival rates, with a median overall survival (OS) of 24.0 days. Treatment with TAS3351 at doses of 20 and 40 mg/kg reduced intracranial luciferase activity and extended the median OS compared to the vehicle-treated group, with statistical significance. Collectively, these findings suggest that TAS3351 has significant antitumor activity against CNS tumors, which is attributed to its remarkable brain penetrability.

### Discussion

In the treatment of patients with NSCLC harboring EGFR-activating mutations, osimertinib is widely recognized as a primary treatment option among EGFR-TKIs and is extensively used as first-line therapy. Osimertinib has demonstrated significant clinical benefits not only in patients with advanced-stage NSCLC but also in those with early-stage disease[57]. Despite its considerable clinical efficacy, osimertinib resistance eventually develops in several patients. A recent study analyzing the mechanisms of resistance to first-line osimertinib treatment using ctDNA revealed no T790M-mediated acquired resistance; instead, the most common resistance mechanisms were MET amplification (17%) and EGFR C797S mutation (6%)[58]. These findings are generally consistent with the previous trends reported in second- and later-line settings; however, the frequency of occurrence of the C797S mutation appears to be lower in patients treated in earlier lines. Previous studies have reported the development of the C797S resistance mutation in 18%–25% of patients in second- or later-line treatment settings[19,20,23]. The difference in the C797S mutation frequency across treatment lines can be partially explained by the biochemical properties of the C797S mutation. As demonstrated in this study, the C797S mutation does not affect the kinetics

of the EGFR enzymatic reaction, as cysteine 797 serves only as a site for covalent binding with osimertinib and is not involved in the catalytic activity of EGFR. In contrast, the T790M mutation increases ATP affinity for EGFR, which will partially explain the observed de novo occurrence of the T790M mutation. Given the biochemical characteristics of the C797S mutation, it is unlikely to spontaneously arise the C797S mutation in the absence of covalent EGFR-TKI treatments. Instead, prolonged osimertinib treatment likely increases the incidence of C797S. Indeed, patients who experience progression-free survival (PFS) longer than the median PFS seem to have a higher incidence of C797S mutations[20,23,58,59]. Therefore, the C797S mutation is expected to emerge in a certain proportion of patients treated with osimertinib or other third-generation EGFR-TKIs, regardless of the treatment lines, including adjuvant settings. Furthermore, longer treatment durations are likely to increase the likelihood of the C797S mutation emergence as a resistance mechanism.

During osimertinib treatment, T790M does not emerge as a resistance mutation; however, subsequent treatment of patients with first- or second-generation EGFR-TKIs will result in the development of the T790M mutation as a resistance mechanism. Although the actual incidence rate of the T790M mutation might be lower, given that de novo T790M mutations are likely to be eradicated by prior osimertinib treatment, the T790M mutation remains a clinical concern as long as first- or second-generation EGFR-TKIs are administered in clinical practice[60]. Moreover, despite the strong intracranial activity of osimertinib, the progression of metastatic lesions within the CNS is often observed[61]. Although further investigation is required to elucidate the resistance mechanisms in intracranial lesions during osimertinib treatment, developing additional treatment options with improved brain penetration and efficacy against concurrent or sole resistant mutations of both T790M and C797S is crucial for later lines of therapy following osimertinib treatment.

Presently, numerous efforts are underway to develop additional treatment options for patients whose tumors have become refractory to or have relapsed after osimertinib treatment. Among these, the combination of amivantamab, an EGFR-MET bispecific antibody with immune cell-directing activity[62], and lazertinib, a CNS-penetrant third-generation EGFR-TKI[63,64], has demonstrated superior efficacy compared to osimertinib as a first-line treatment for EGFR-mutated advanced NSCLC[65]. Although the primary mechanisms of amivantamab resistance are not yet fully elucidated, patients treated with lazertinib develop the C797S mutation as a resistance mutation[66]. Antibody drug conjugates (ADCs), which comprise monoclonal antibodies conjugated to cytotoxic payloads via a chemical linker,

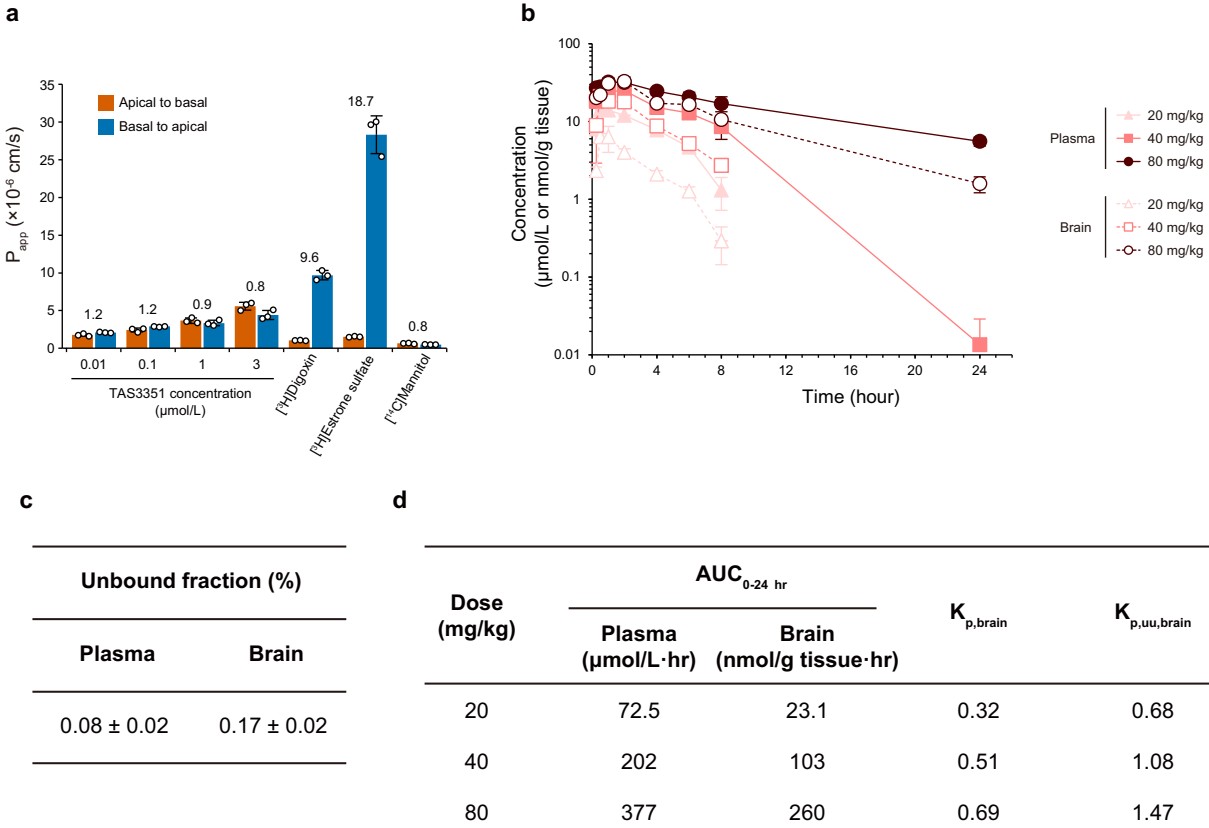

**Fig. 7 | Permeability of TAS3351 across Caco-2 monolayers and pharmacokinetics of TAS3351 in the plasma and brain in nude mouse. a** Transcellular transport of TAS3351 and reference compounds across Caco-2 cell monolayers was evaluated. The transport of TAS3351 was assessed from both apical to basal and basal to apical in concentrations of 0.01–3 μmol/L. For reference, 1 μmol/L [$^3$H] digoxin and 0.1 μmol/L [$^3$H]estrone sulfate were evaluated for P-gp and BCRP transporter-mediated efflux, respectively. In addition, 10 μmol/L [$^{14}$C]mannitol was used to assess paracellular permeability. The bars indicate the mean $P_{app}$ values and standard deviations (s.d.) from triplicate experiments. Individual values are shown as dots. The values above each bar graph for the tested compounds represent the $P_{app}$ ratio, which is calculated by correcting the mean values of basal to apical $P_{app}$ compared to those of apical to basal $P_{app}$. **b** The plasma and brain concentration-time profiles of TAS3351 in nude mice (BALB/cAJcl-nu/nu) following a single oral administration are illustrated. The plasma and brain samples were collected at 0.25, 0.5, 1, 2, 4, 6, 8, and 24 h after administration in three mice. The mean values and s.d. from three animals are shown. The TAS3351 concentration in the plasma (20 mg/kg group) and brain (20 and 40 mg/kg groups) at the 24-h time point was below the lower limit of quantification (BLQ). Total 72 mice were used. **c** Unbound fractions in the plasma and brain, which were determined using Rapid Equilibrium Dialysis device system spiked with TAS3351 at 0.5 μmol/L for the plasma and 20 nmol/g for the brain; data are presented as the percent of mean ± s.d. (*n* = 3). **d** The $AUC_{0-24\ hr}$ values of TAS3351 are calculated from the data described in (**b**). $K_{p,brain}$ is obtained by dividing brain $AUC_{0-24\ hr}$ with plasma $AUC_{0-24\ hr}$. Unbound $AUC_{0-24\ hr}$ is calculated by multiplying $AUC_{0-24\ hr}$ by each unbound fraction described in (**c**). $K_{p,uu,brain}$ is obtained by dividing unbound brain AUC with unbound plasma AUC.

have emerged as crucial treatment options for patients with cancer. In EGFR-mutated NSCLC, patritumab deruxtecan, a HER3-directed ADC with a topoisomerase I inhibitor payload, has shown clinical activity in EGFR TKI-resistant NSCLC, irrespective of the resistance mechanisms, including the C797S mutation[67]. A phase 3 trial evaluating patritumab deruxtecan in patients with advanced EGFR-mutated NSCLC who previously received EGFR-TKI treatment is currently underway (NCT05338970). These next-generation therapeutic strategies, which are distinct from traditional EGFR-TKIs, are expected to reshape the treatment landscape for EGFR-mutated NSCLC. However, fourth-generation EGFR-TKIs such as TAS3351 may still offer significant advantages over these therapeutics with respect to oral administration and brain penetrability. TAS3351 is not a substrate of P-gp and BCRP and has shown remarkably higher $K_{p,uu,brain}$ values (0.68–1.47) in mice compared to other presently available EGFR-TKIs, such as osimertinib[32,68], which has demonstrated clinical responses and antitumor effect in an intracranial mouse transplant model. These data suggest that TAS3351 will achieve good brain exposure and demonstrate robust efficacy in the brain in clinical settings.

In the present study, we demonstrate that TAS3351 overcomes concurrent or sole resistant mutations of the T790M and C797S, while sparing wild-type EGFR activity through a non-covalent mechanism, and exhibits

superior brain penetrability, consistent with its classification as a fourth-generation EGFR-TKI. Besides, TAS3351 features a unique pyrrolopyrimidine-quinoline core structure, which is structurally distinct from brigatinib and its derivatives, other fourth-generation EGFR-TKIs, such as BDTX-1535, BLU-945, and BI-4020, and allosteric EGFR inhibitors, such as JBJ-09-063. This distinct structure offers the potential for a differentiated spectrum of inhibitory potency and resistance mechanisms. On the other hand, we also demonstrate that TAS3351 exhibits an obvious difference in inhibitory potency between the ex19del and L858R EGFR mutation, as similarly seen in the first-generation EGFR-TKIs. However, the detailed mechanisms underlying this relatively large difference in TAS3351 remain unclear. Since patients with tumors harboring the EGFR L858R mutation often demonstrate inferior clinical outcomes compared to those with the EGFR ex19del mutation, the relative lack of inhibition against the EGFR L858R mutation in TAS3351 could potentially limit its clinical application.

At present, no fourth-generation EGFR-TKIs have been approved, although several candidates have been evaluated in the clinical setting. Among them, BDTX-1535 is the most advanced agent, presently undergoing a phase 2 clinical trial (NCT05256290), with a partial response observed in patients with NSCLC harboring the C797S mutation[69]. Given the broad spectrum of inhibition of various osimertinib-resistant mutations,

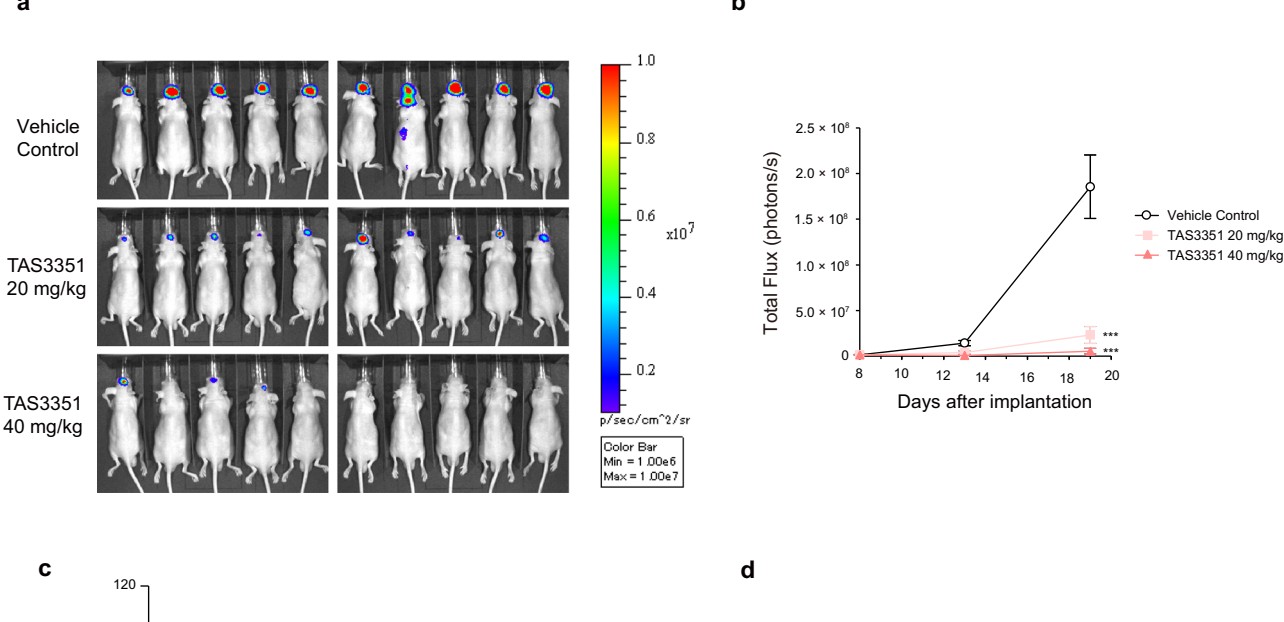

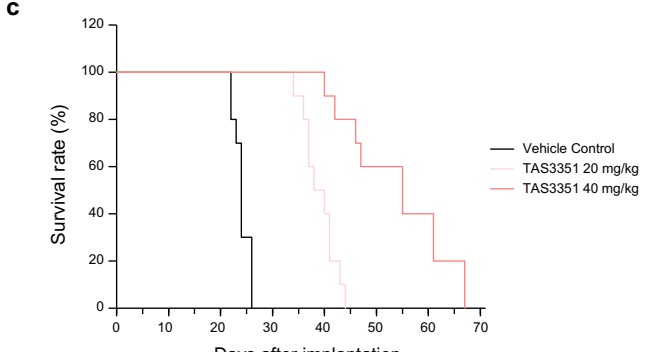

**Fig. 8 | Antitumor activity of TAS3351 in intracranially transplanted mouse allograft model.** Nude mice (BALB/cAJcl-nu/nu) were intracranially transplanted with NIH/3T3 cells, which stably expressed human EGFR harboring ex19del/T790M/C797S and luciferase. Mice bearing intracranial allografts were divided into three groups of ten mice on day 8 and treated with TAS3351 from day 9 until the day before the final evaluation of survival time. Mice in the vehicle control group were treated with a vehicle of TAS3351 (20% HP-β-CD, 0.1 mol/L HCl) **a** Individual bioluminescence images of the mice on day 19. 150 mg/kg luciferin was administrated intraperitoneally, and bioluminescence images were acquired using IVIS Lumina II imaging system. **b** The mean total flux and standard error of the mean (s.e.m.) from ten mice at day 8, 13, and 19 after transplantation are graphed. Dunnett's test is used to compare the logarithmically transformed total flux of each treated group on Day 19 with that of the control group. ***$p < 0.0001$ **c** Survival curve of the mice in each group are presented. **d** The median overall survival (OS) and $p$-values are summarized. The log-rank test is used to compare the survival curve of each TAS3351 group with that of the vehicle control group.

including the C797S, and significant free drug exposure in the brain[69], further detailed reports on the clinical outcomes of BDTX-1535 in patients with the C797S mutation are awaited. In summary, the preclinical findings described herein provide a strong scientific rationale for the clinical evaluation of TAS3351 in patients with EGFR-mutated NSCLC.

## Data availability

The crystallographic data generated in this study have been deposited in Research Collaboratory for Structural Bioinformatics Protein Data Bank (RCSB PDB) with the accession codes (9KL4 and 9KLW). The crystallographic data and refinement statistics are available in the Supplementary Information file as Supplementary Tables 1 and 2. The coordinate Crystallographic Information Files for 9KL4 and 9KLW are found in Supplementary Data 1 and 3, respectively. The structure-factor Crystallographic Information Files for 9KL4 and 9KLW are found in Supplementary Data 2 and 4, respectively. All numerical results underlying the graphs and table presented in this study are available in Supplementary Data 5–8. The uncropped immunoblotting data with molecular markers for Figs. 6a, b are available in the Supplementary Information file as Supplementary Fig. 6 and 7, respectively. All additional relevant data and information in this study are available from corresponding author upon request.

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

## Acknowledgements

The authors thank Drs. Teruhiro Utsugi, Takeshi Sagara, Yoshikazu Iwasawa, Kazuhiko Yonekura, and Kazutaka Miyadera for mentoring this work. We would also like to thank all those who contributed to this work at the Discovery and Preclinical Research Division of Taiho Pharmaceutical Co., Ltd. We also thank the MD Anderson Cancer Center for their invaluable support in conducting this research. PC-9 and Ba/F3 cells were provided by the RIKEN BRC through the National Bio-Resource Project of the MEXT, Japan. This study was funded by Taiho Pharmaceutical Co., Ltd., a wholly owned subsidiary of Otsuka Holdings Co., Ltd.

## Author contributions

H.K., F.Y., and S.M. conceived, designed, and supervised the entire experiment. H.K., F.Y., Y.K., F.Y., S.T., R.M., and T.S. implemented experiments, acquired data, and all authors participated in data interpretation. H.K. wrote the manuscript, and all authors reviewed the manuscript.

## Competing interests

All authors are employees of Taiho Pharmaceutical Co., Ltd., a wholly owned subsidiary of Otsuka Holdings Co., Ltd.
