## [Transparent Peer Review file · Communications Medicine]

TAS3351 is a brain penetrable EGFR-TKI that overcomes T790M and C797S resistant mutations

Corresponding Author: Dr Hidefumi Kasuga

Version 0:

Reviewer comments:

Reviewer #1

(Remarks to the Author)

In the present study, Kasuga et al. demonstrated the discovery of TAS3351, a novel fourth-generation EGFR TKI. In general, this is a comprehensive investigation from structure, in vitro, in vivo assay. However, several issues need to be further addressed.

1. With the osimertinib and FLAURA2 treatment strategy as the first-line treatment for EGFR-mutant NSCLC, acquired C797S without T790M, rather than concurrent EGFR T790M and C797S remains as a major on-target resistance mechanism in clinics. Thus, the authors should emphasize the data of TAS3351 in this subset of setting in each assay.
2. The author should present downstream protein expression and phosphorylation data including pAkt rather than only EGFR and ERK to show the data of downstream inhibitor.
3. In cell viability assay, the authors should further construct human cell line model with EGFR 21L858R and C797S mutation, rather than only PC9(C797S), as the BaF3 cell line model demonstrated difference in EGFR DEL19/C797S and EGFR 21L858R/C797S by TAS3351 treatment.

Reviewer #2

(Remarks to the Author)

The current study demonstrated the superiority of the fourth generation EGFR-TKI TAS3351 in EGFR-mutated NSCLC especially harboring C797S compared to other EGFR-TKIs. As described in discussion, it is conceivable that C797S could remain a cause of EGFR-TKI resistance even after amivantamab becomes available for first-line treatment. However, the possibility that amivantamab treatment may affect the frequency of the secondary C797S mutation is undeniable and further reports are awaited. Another interesting aspect of this study is that it suggests that TAS3351 may be highly effective against brain metastases. On the other hand, TAS is a reversible EGFR-TKI and not an irreversible TKI like osimertinib or BDTX-1535. TAS3351 is expected to have good blood brain barrier permeability, but whether it is as effective as osimertinib against brain metastases in humans awaits further evaluation in clinical trials. This paper is multifaceted, including biochemical kinetics assays, 3D structural analysis, and transporter analysis, and I think the results are sufficiently robust. The development of the new fourth-generation EGFR-TKI TAS3351 is of great clinical value, and the publication of the present paper in this journal is significant. Some minor comments are noted below.

- In line 49, "In addition, the T790M mutation increases the affinity of EGFR for ATP, thereby attenuating the effectiveness of ATP competitive EGFR-TKIs compared to EGFR harboring the ex19del or L858R mutations". You can list reference paper for this sentence.
- Figure 3a, the cell proliferation curves are probably the same between BaF3 Ex19/T790M and BaF3 Ex19/C797S. Please check for any errors.
- Are PC9 and PC9 (C797S) swapped in the osimertinib treatment group in Fig. 4b?
- Other EGFR mutations such as L718Q and G718V have also been reported to be involved in osimertinib resistance. If you are investigating the effect of TAS3351 on other cells harboring other osimertinib-resistant mutations, please include the results if available.
- Fig6 shows a weakened inhibitory effect of TAS3351 on activation of EGFR and ERK at 24 hours after a TAS3351 exposure. I wonder if a divided dosing of TAS3351 would enhance the anti-tumor effect in vivo? Please provide your evaluation if you have done in such a way.

Version 1:

Reviewer comments:

Reviewer #1

(Remarks to the Author)

In the revised manuscript, the authors have addressed most of the review comments. The only one I suggest to be further clarified is the obvious difference of TAS3531 between EGFR Del19 and 21L858R mutation, as presented in the suppl. Fig5 and Fig 6.

The relative lack of inhibition in EGFR 21L858R mutation is clinically important as patients acquired C797S with or without T790M but the sensitizing mutation (Del19/21L858R) remained. As such, I suggest the differences should be objectively discussed as the potential limitation in the discussion part rather than only in line 157.

Reviewer #2

(Remarks to the Author)

I have no further concerns regarding this paper.

Tracking number: COMMSMED-25-1011A

Corresponding author: Hidefumi Kasuga

Title: Discovery of TAS3351, a brain-penetrable fourth-generation EGFR-TKI that overcomes T790M and C797S resistance mutations

Reviewer #1 (Remarks to the Author):

In the present study, Kasuga et al. demonstrated the discovery of TAS3351, a novel fourth-generation EGFR TKI. In general, this is a comprehensive investigation from structure, in vitro, in vivo assay. However, several issues need to be further addressed.

1. With the osimertinib and FLAURA2 treatment strategy as the first-line treatment for EGFR-mutant NSCLC, acquired C797S without T790M, rather than concurrent EGFR T790M and C797S remains as a major on-target resistance mechanism in clinics. Thus, the authors should emphasize the data of TAS3351 in this subset of setting in each assay.

- Thank you for the important comment that reflect the current clinical practice. Following the comment, we added the sentence in line of 244, in addition to the modification in lines of 181, 231, 351, and 375.

2. The author should present downstream protein expression and phosphorylation data including pAkt rather than only EGFR and ERK to show the data of downstream inhibitor.

- We respectfully agree with your point. Indeed, we initially had planned to evaluate pAKT and total AKT, as well as pEGFR and β -actin. However, due to a technical problem, the total AKT data were omitted (part of the immune blotting signal was accidentally separated when the PVDF membrane was cut according to the molecular weight marker), although the phosphorylated Akt data were deemed valid. Following your suggestion, we performed again an immune blotting analysis of total Akt using frozen Western blot samples stored in a deep freezer at -80°C and successfully obtained total Akt data. As such, we have revised Figure 6 to include the pAkt and total Akt data, as well as the manuscripts.

3. In cell viability assay, the authors should further construct human cell line model with EGFR 21L858R and C797S mutation, rather than only PC9(C797S), as the BaF3 cell line model demonstrated difference in EGFR DEL19/C797S and EGFR 21L858R/C797S by TAS3351 treatment.

- We appreciate your valuable comment regarding the need to evaluate the human cell line model harboring EGFR with L858R and C797S. At the time we initiated this work, NCI-H1975 was almost the only publicly available model harboring L858R mutation. Therefore, we started with NCI-H1975, aiming to establish a human cell line introduced with C797S in EGFR locus,

utilizing CRISPR-Cas9-based gene editing. Nevertheless, the multiple attempts to make NCI-H1975 harboring C797S mutation resulted in failure. We then pivoted our focus to PC-9 and successfully established the PC-9 harboring C797S. We haven't elucidated the reason of the failure, however reagents we utilized and the experiment conditions may have affected the efficacy of gene editing in NCI-H1975 cell line. Simultaneously, we had searched for the PDX models harboring both L858R and C797S mutation, with or without the T790M mutation. However, we could not have found those models at an appropriate timing. Therefore, we deprioritized our efforts to make a human cell line harboring L858R and C797S and to evaluate in PDX, instead, we decided to expand our analysis to include a comprehensive set of EGFR biochemical analysis and in vitro phosphor-EGFR evaluation, over a full set of mutation variations. We recognize that there is a substantial difference of inhibitory potency in TAS3351 between ex19del and L858R, however, the difference is likely due to the different affinity of TAS3351 toward ex19del (Ki: 4.80 pmol/L) and L858R (Ki: 259 pmol/L), although the reason for this difference remains unclear. With all due respect, to establish a human cell line harboring L858R and C797S mutation will not directly contribute to reveal the mechanism by which TAS3351 exhibits distinguished inhibitory potency against the ex19del and L858R mutation. Rather, further structural and biochemical analysis between ex19del and L858R would provide additional insight into this. We believe that these further analyses are beyond the scope of this study, and would like to address the mechanisms of largely distinguished affinity between ex19del and L858R in future studies. Hence, we added additional languages have been added in the line of 157. We sincerely hope you appreciate our thought.

Reviewer #2 (Remarks to the Author):

The current study demonstrated the superiority of the fourth generation EGFR-TKI TAS3351 in EGFR-mutated NSCLC especially harboring C797S compared to other EGFR-TKIs. As described in discussion, it is conceivable that C797S could remain a cause of EGFR-TKI resistance even after amivantamab becomes available for first-line treatment. However, the possibility that amivantamab treatment may affect the frequency of the secondary C797S mutation is undeniable and further reports are awaited. Another interesting aspect of this study is that it suggests that TAS3351 may be highly effective against brain metastases. On the other hand, TAS is a reversible EGFR-TKI and not an irreversible TKI like osimertinib or BDTX-1535. TAS3351 is expected to have good blood brain barrier permeability, but whether it is as effective as osimertinib against brain metastases in humans awaits further evaluation in clinical trials. This paper is multifaceted, including biochemical kinetics assays, 3D structural analysis, and transporter analysis, and I think the results are sufficiently robust. The development of the new fourth-generation EGFR-TKI TAS3351 is of great clinical value, and the publication of the present paper in this journal is significant.

Some minor comments are noted below.

- In line 49, “In addition, the T790M mutation increases the affinity of EGFR for ATP, thereby attenuating the effectiveness of ATP competitive EGFR-TKIs compared to EGFR harboring the ex19del or L858R mutations”. You can list reference paper for this sentence.

- We would like to express our sincere apologies for the oversight in failing to cite the fundamental report regarding the T790M characterization. The following publication was added in the line 51: *Yun et al., The T790M mutation in EGFR kinase causes drug resistance by increasing the affinity for ATP. PNAS 105, 2070-2075 (2008)*

- Figure 3a, the cell proliferation curves are probably the same between BaF3 Ex19/T790M and BaF3 Ex19/C797S. Please check for any errors.

- I would like to express my gratitude for the thorough review of the data. We have once again verified that the BaF3 ex19del/T790M and ex19del/C797S references are appropriate, despite the similarity of the data. For your reference, please find below the row data used for analysis:

BaF3-EGFR (ex19del/C797S) – mean of T/C (%)

Conc. (nM)	10,000	3,000	1,000	300	100	30	10	3	1	0.3
TAS3351	1.2699	4.6249	11.2531	13.7482	14.2847	17.4437	35.2792	70.4661	91.5653	98.7732
Gefitinib	0.4014	6.8998	10.9610	13.7937	15.7235	24.6058	62.5321	88.9339	103.9472	107.3514
Erlotinib	0.8978	5.7280	11.5103	12.0141	14.3616	22.1435	52.3329	81.9214	96.9783	103.1961
Afatinib	0.0171	-0.0554	0.2233	5.0227	11.1225	22.8838	61.4603	85.6049	99.8457	104.6844
Osimertinib	-0.0415	-0.0563	8.7896	66.8901	91.8967	100.0808	102.5534	101.1664	101.9028	102.0882

BaF3-EGFR (ex19del/C797S) – SD (%)

Conc. (nM)	10,000	3,000	1,000	300	100	30	10	3	1	0.3
TAS3351	0.2146	0.4166	3.1945	2.8212	2.0048	4.6371	10.1531	7.8579	4.2203	0.4323
Gefitinib	0.0454	0.8764	1.1857	2.1644	2.2093	2.2568	7.3966	5.0222	3.3435	2.3368
Erlotinib	0.3461	0.2135	1.2976	1.6767	2.2918	1.6509	7.9218	3.8457	1.7935	1.6937
Afatinib	0.0658	0.0051	0.0894	0.9792	1.4237	1.6116	6.9637	3.8926	0.6377	2.8531
Osimertinib	0.0033	0.0087	8.4671	4.5549	2.0465	1.6242	2.6381	1.1007	0.8996	1.4589

BaF3-EGFR (ex19del/T790M) – mean of T/C (%)

Conc. (nM)	10,000	3,000	1,000	300	100	30	10	3	1	0.3
TAS3351	1.2825	2.3642	2.2983	2.0887	3.8258	20.3298	51.8195	85.4918	97.0697	103.1360
Gefitinib	1.3403	49.7555	97.7541	106.9829	106.6586	106.5219	104.3474	106.1935	103.8274	104.4688

Erlotinib	4.0385	38.2088	83.5438	99.6854	103.9842	104.0042	102.1957	104.2249	104.9730	104.1148
Afatinib	-0.0273	-0.0407	0.0912	8.1414	82.4343	124.3089	147.3514	131.7615	115.3165	109.4522
Osimertinib	-0.0803	-0.0900	-0.0674	0.6818	1.1783	3.0813	15.9020	91.0655	102.1690	106.5625

BaF3-EGFR (ex19del/T790M) – SD (%)

Conc. (nM)	10,000	3,000	1,000	300	100	30	10	3	1	0.3
TAS3351	0.4715	0.7839	0.6457	0.7388	2.8137	19.8204	15.4792	11.0784	5.5239	2.7099
Gefitinib	0.8017	3.4303	3.3374	2.9143	3.1439	3.1988	3.7412	0.6465	2.0061	1.3029
Erlotinib	2.3279	8.0342	5.2594	1.5024	1.4881	1.6283	1.6176	1.5663	2.0091	2.0513
Afatinib	0.0595	0.0098	0.0884	4.0289	1.8352	3.4716	10.0032	8.2559	2.9343	3.4398
Osimertinib	0.0047	0.0063	0.0159	0.2247	0.5428	1.4741	4.3686	1.9590	0.0943	1.1644

- Are PC9 and PC9 (C797S) swapped in the osimertinib treatment group in Fig. 4b?

- Our sincere apologies that we mistakenly swapped the labels of PC-9 and PC-9 (C797S) in Fig. 4b. We corrected the labels appropriately.

- Other EGFR mutations such as L718Q and G718V have also been reported to be involved in osimertinib resistance. If you are investigating the effect of TAS3351 on other cells harboring other osimertinib-resistant mutations, please include the results if available.

- We have completed a full analysis of TAS3351's inhibitory spectrum in relation to a range of broad resistant mutations, including L718Q and L718V. The data indicate that TAS3351 is not effective in cases of EGFR harboring L718Q and L718V, respectively. However, it displays a unique inhibitory spectrum when compared with osimertinib. We intend to publish these findings in the future. Therefore, we would be grateful if you could agree not to mention these findings in this report.

- Fig6 shows a weakened inhibitory effect of TAS3351 on activation of EGFR and ERK at 24 hours after a TAS3351 exposure. I wonder if a divided dosing of TAS3351 would enhance the anti-tumor effect in vivo? Please provide your evaluation if you have done in such a way.

- We would like to express our gratitude for your insightful comments. We previously conducted an exploratory analysis of QD and BID dosing regimen comparison, administering equivalent amounts of TAS3351 daily in the NIH/3T3-EGFR (ex19del/T790M/C797S) model. However, this study did not demonstrate any enhancement in efficacy. That being said, we have not conducted a detailed analysis of PD

marker evaluation between QD and BID, and we only analyzed in one model. As evaluating split dosing so that more continuous PD marker inhibition is achieved would be of great interest, we added additional comments in the line 279. We are in agreement that the administration of TAS3351 at multiple doses per day has the potential to contribute to the continuous inhibition of phosphor-EGFR, which may result in an enhancement in efficacy.

10 January 2026

Tracking number: COMMSMED-25-1011B

Corresponding author: Hidefumi Kasuga

Title: Discovery of TAS3351, a brain-penetrable fourth-generation EGFR-TKI that overcomes T790M and C797S resistance mutations

Reviewer #1 (Remarks to the Author):

In the revised manuscript, the authors have addressed most of the review comments. The only one I suggest to be further clarified is the obvious difference of TAS3531 between EGFR Del19 and 21L858R mutation, as presented in the suppl. Fig5 and Fig 6.

The relative lack of inhibition in EGFR 21L858R mutation is clinically important as patients acquired C797S with or without T790M but the sensitizing mutation (Del19/21L858R) remained. As such, I suggest the differences should be objectively discussed as the potential limitation in the discussion part rather than only in line 157.

We again deeply acknowledge the insightful suggestion regarding the difference in inhibitory potency in TAS3351 against ex19del and L858 mutation. According to this suggestion, we have added the comments in the discussion part regarding the difference and its potential limitation in the clinical setting.

Reviewer #2 (Remarks to the Author):

I have no further concerns regarding this paper.